# MoE-Gyro: Self-Supervised Over-Range Reconstruction and Denoising for MEMS Gyroscopes

**Feiyang Pan**[1†], **Shenghe Zheng**[2†], **Chunyan Yin**[1*], **Guangbin Dou**[1*]

[1] Southeast University      [2] Harbin Institute of Technology

230238437@seu.edu.cn

## Abstract

MEMS gyroscopes play a critical role in inertial navigation and motion control applications but typically suffer from a fundamental trade-off between measurement range and noise performance. Existing hardware-based solutions aimed at mitigating this issue introduce additional complexity, cost, and scalability challenges. Deep-learning methods primarily focus on noise reduction and typically require precisely aligned ground-truth signals, making them difficult to deploy in practical scenarios and leaving the fundamental trade-off unresolved. To address these challenges, we introduce **Mixture of Experts for MEMS Gyroscopes (MoE-Gyro)**, a novel self-supervised framework specifically designed for simultaneous over-range signal reconstruction and noise suppression. **MoE-Gyro** employs two experts: an Over-Range Reconstruction Expert (ORE), featuring a Gaussian-Decay Attention mechanism for reconstructing saturated segments; and a Denoise Expert (DE), utilizing dual-branch complementary masking combined with FFT-guided augmentation for robust noise reduction. A lightweight gating module dynamically routes input segments to the appropriate expert. Furthermore, existing evaluation lack a comprehensive standard for assessing multi-dimensional signal enhancement. To bridge this gap, we introduce **IMU Signal Enhancement Benchmark (ISEBench)**, an open-source benchmarking platform comprising the GyroPeak-100 dataset and a unified evaluation of IMU signal enhancement methods. We evaluate **MoE-Gyro** using our proposed **ISEBench**, demonstrating that our framework significantly extends the measurable range from ±450°/s to ±1500°/s, reduces Bias Instability by 98.4%, and achieves state-of-the-art performance, effectively addressing the long-standing trade-off in inertial sensing. Our code is available at: https://github.com/2002-Pan/Moe-Gyro

## 1 Introduction

MEMS gyroscopes are essential inertial sensors extensively utilized in navigation and control systems such as autonomous vehicles, unmanned aerial vehicles (UAVs), robotics, and precision-guided munitions[1, 2, 3]. In these high-dynamic applications, critical performance metrics of gyroscopes include measurement range (full-scale angular velocity) and noise characteristics, notably Angle Random Walk (ARW) and Bias Instability (BI). However, commercial MEMS gyroscopes typically encounter a fundamental performance trade-off: enhancing the measurement range generally results in elevated ARW and BI, whereas sensors optimized for low noise inherently possess a restricted angular velocity measurement capability[4, 5]. This fundamental contradiction significantly limits their effectiveness in high-angular-rate scenarios requiring precise inertial measurements.

Addressing this critical limitation without incurring additional sensor complexity or manufacturing costs remains a significant and unresolved challenge in inertial sensor research. Traditional solutions

---

*Corresponding Author(yincy@seu.edu.cn; gdou@seu.edu.cn). †Equal Contribution.

39th Conference on Neural Information Processing Systems (NeurIPS 2025).

primarily involve structural and circuit-level strategies, such as resonant frequency tuning (mode-splitting)[6, 7], closed-loop force-feedback control[8, 9], and multi-range readout electronics[10]. Although these methods deliver incremental gains, they require tighter fabrication tolerances and more complex control circuitry, increasing both power draw and manufacturing cost [11]. Thus, traditional strategies have yet to adequately resolve these critical trade-offs[12]. Recent advancements in deep learning have emerged as promising solutions for mitigating noise in gyroscopes and, by extension, in complete inertial-measurement-unit (IMU) signals (e.g., CNN[13], LSTM_GRU[14], HEROS_GAN[15]). However, these approaches rely on fully supervised training and thus require precisely time-synchronized noisy/clean rotation pairs data that are expensive to collect. Meanwhile, current self-supervised methods (e.g., LIMU-BERT[16], IMUDB[17]) lack a unified framework that handles both denoising and over-range reconstruction, leaving unresolved the core trade-off that low-noise sensors accept only a limited angular rate range. Moreover, previous work is limited to a few single-signal test environments, lacking a multidimensional benchmark that captures the full spectrum of enhancement performance; a public suite spanning diverse operational scenarios is essential for fair comparison and real progress.

To overcome the limitations above, we introduce **MoE-Gyro**, the first self-supervised unified architecture that tackles both over-range reconstruction and denoising. A lightweight gate dynamically routes each input signal segment to two specialized experts, an Over-Range Reconstruction Expert (ORE) and a Denoise Expert (DE), thus cutting inference memory because, in practice, only a single expert is active for most segments. Both experts are trained end-to-end on a shared Masked Autoencoder(MAE) [18] backbone with purely self-supervised objectives, eliminating the need for costly, time-synchronised ground-truth labels. We further introduce task-specific more optimisations. For the ORE, a Gaussian-Decay Attention (GD-Attn) module in the decoder automatically focuses on the most relevant context for peak reconstruction, while a physics-informed energy regulariser (PINN) enforces consistency with the gyroscope's mechanical model, boosting generalisation across sensors. For the DE, we adopt a dual-branch complementary cross-mask that captures weak signal features while smoothing high-frequency noise, and we employ FFT-guided noise injection during training to strengthen the learned denoising mapping. Together, these innovations deliver a unified, fully self-supervised solution that simultaneously broadens range and suppresses noise in commercial MEMS gyroscopes. In addition, we release **ISEBench**, the first open-source benchmark with a unified suite of evaluation metrics, providing a common benchmark for future research on IMU signal enhancement. Our key contributions can be summarized as follows:

• Unified self-supervised MoE framework that simultaneously reconstructs over-range signals and reduces noise, breaking the long-standing range–noise trade-off without extra hardware.

• We propose a Gaussian-Decay Attention (GD-Attn) and a physics-informed neural network (PINN) loss, extending the measurable range of a typical $\pm450$ °/s MEMS gyroscope to $\pm1500$ °/s.

• We design a dual-branch complementary masking strategy combined with FFT-guided augmentation, significantly reducing Bias Instability on the test set by 98.4% .

• We release **ISEBench**, the first open source benchmark specifically tailored for comprehensive evaluation of IMU signal enhancement, along with a dedicated dataset for over-range reconstruction, facilitating fair comparisons and fostering rapid progress within the community.

## 2 Related Works

### 2.1 IMU over-range signal reconstruction.

Reconstructing saturated signal segments in IMUs remains a critical but significantly under explored problem. Among the few representative studies, HEROS-GAN[15] formulates the problem as a fully supervised generative task, relying heavily on paired saturated and reference data for training. Alternatively, Matlab 2023b[19] release introduced a polynomial-based extrapolation function that estimates saturated peaks from their neighboring points; however, this model-driven approach is inherently sensitive to noise and struggles under highly dynamic conditions. However, existing studies have not explored self-supervised approaches or integrated IMU-specific physical constraints into the reconstruction task.

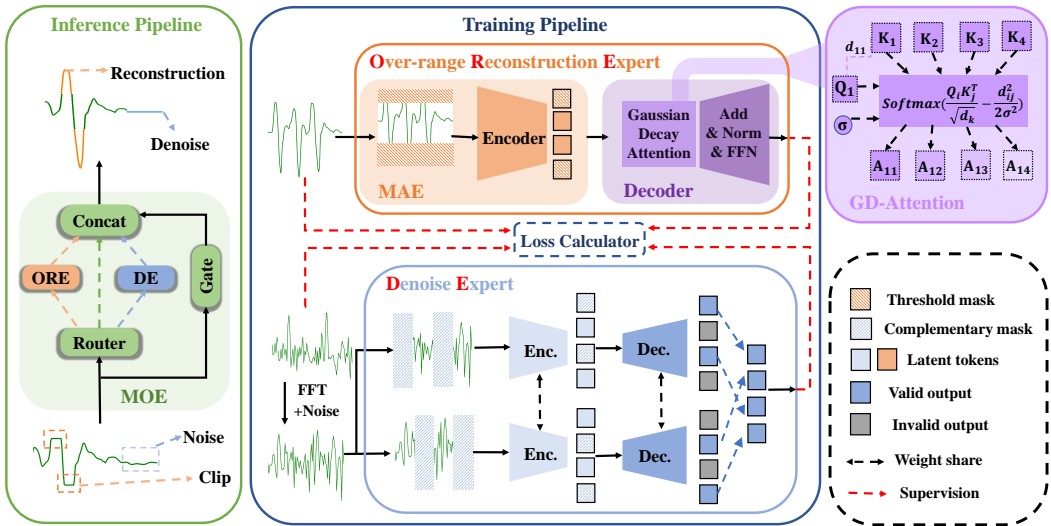

Figure 1: Pipeline of MoE-Gyro framework. During inference, a low-quality signal stream is segmented, routed by a gate to suitable expert, and the enhanced outputs are concatenated to form the final signal. During training, both experts are optimised in a fully self-supervised manner on a shared MAE backbone, each equipped with task-specific masking, attention, and loss mechanisms.

## 2.2 IMU signal denoise.

In contrast to the sparse literature on over-range reconstruction, signal enhancement research for IMUs has largely centred on noise suppression. Classical, model-driven filters such as EMD denoising[20] and Savitzky–Golay smoothing[21] can isolate and attenuate noise, yet they depend on accurate noise priors, which often fail to transfer across sensors or operating conditions. More recent data-driven approaches, including CNN[13], LSTM-GRU hybrids[14], and the IMUDB[17], replace explicit priors with learned representations and have therefore attracted wider adoption.

## 3 Method

**Motivation.** Although recent supervised and self-supervised models have markedly improved IMU denoising, simply grafting multiple task heads onto one backbone to unify denoising and over-range reconstruction proves unreliable. After normalization, clipped-peak errors dwarf background noise by several orders of magnitude, so reconstruction gradients dominate training, and the network largely ignores the denoising head. The resulting scale-imbalance leaves one task under-fitted and the other only partially solved, curbing overall accuracy and generalization[22, 23, 24].

This observation motivates a decoupled, self-supervised Mixture-of-Experts design in which each task is handled by a dedicated specialist while global features are shared only when beneficial. By adopting a shared Masked Autoencoder (MAE) encoder combined with lightweight task-specific decoders, each expert is allowed to specialize independently. Moreover, we introduce dedicated task-oriented masking strategies, a Gaussian-Decay Attention mechanism, and physics-informed constraints to ensure stable and targeted optimization. This modular, decoupled design ultimately enables more effective training and significantly improved enhancement performance.

**Overview.** Figure 1 shows the **MoE-Gyro** architecture for self-supervised inertial signal enhancement. The raw signal is first segmented, and a gate then applies a simple heuristic to route each segment to the Over-Range Reconstruction Expert (ORE), the Denoise Expert (DE), both experts, or directly to the output. The ORE follows a standard MAE backbone but adds a task-specific threshold mask to focus on in-range features, and inserts a Gaussian-Decay Attention block in the decoder to selectively amplify peak region information during reconstruction. The DE adopts a dual-branch MAE design in which two parameter-shared encoders/decoders operate on complementary masks, allowing the network to capture intrinsic signal correlations while aggressively suppressing high-

frequency noise. Finally, the gated outputs of the two experts are concatenated to yield the enhanced signal. The precise routing and concating logic is summarized in Algorithm 1.

### 3.1 Over-range Reconstruction Expert

**Gaussian-Decay Attention.** Windowed or local attention has proved effective in vision and language models because it reduces distraction from distant, less relevant context[25, 26]. Such locality is especially valuable for over-range reconstruction, because the information required to restore a clipped peak is concentrated within a short temporal window around the peak. However, a fixed window ignores sensor-specific dynamics and, being a non-differentiable mask, cannot adapt itself during learning. Inspired by these issues, we introduce **Gaussian-Decay Attention (GD-Attn)**, which replaces the binary window mask with a learnable, continuous Gaussian bias. For a query–key pair $(i, j)$ separated by $d_{ij} = |i - j|$ steps, GD-Attn adds a learnable Gaussian bias $B_{ij} = -\frac{d_{ij}^2}{2\sigma^2}$, where the single trainable parameter $\sigma$ is initialised

---

**Algorithm 1:** MoE route and concat

**Input:** segment $x[1{:}L]$
**Output:** enhanced $y[1{:}L]$
1  $y \leftarrow x$;
2  $peak \leftarrow$ 3 consecutive clipped?;
3  $noise \leftarrow$ run of $n$ samples $< \tau$?;
4  **if** $peak$ **then**
5  $\quad$ $\hat{p} \leftarrow \text{PeakExpert}(x)$
6  **end**
7  **if** $noise$ **then**
8  $\quad$ $\hat{n} \leftarrow \text{DenoiseExpert}(x)$
9  **end**
10 **for** $t \leftarrow 1$ **to** $L$ **do**
11 $\quad$ **if** $peak$ **and** $x_t$ clipped **then**
12 $\quad\quad$ $y_t \leftarrow \hat{p}_t$;
13 $\quad$ **else if** $noise$ **and** $|x_{t:t+n-1}| < \tau$ **then**
14 $\quad\quad$ $y_{t:t+n-1} \leftarrow \hat{n}_{t:t+n-1}$; $t \leftarrow t + n$;
15 $\quad$ **end**
16 **end**
17 **return** $y$

---

to a nominal window size and clamped for stability. With queries $Q$, keys $K$, and values $V$, the resulting attention is

$$\text{Output} = \tilde{A}V, \qquad \tilde{A} = \text{softmax}\big(QK^\top / \sqrt{d_k} + B\big) \tag{1}$$

This Gaussian bias yields a soft, differentiable window whose effective width is learned end-to-end; as $\sigma \to \infty$ the bias disappears and GD-Attn reduces to standard global attention, whereas finite $\sigma$ smoothly down-weights distant tokens and concentrates capacity on the peak region.

**Correlation Loss.** Pure $L_2$ reconstruction matches amplitudes but overlooks local dynamics, often smoothing peaks. To recover both trend and extrema we define a two-term *correlation loss* $\mathcal{L}_{\text{corr}}$:

$$\mathcal{L}_{\text{corr}} = \frac{1}{|\mathcal{M}|} \sum_{t \in \mathcal{M}} (\Delta x_t - \Delta \hat{x}_t)^2 \; + \; \lambda_{\text{sign}} \frac{1}{|\mathcal{E}|} \sum_{t \in \mathcal{E}} (x_t - \hat{x}_t)^2 \tag{2}$$

where $\Delta x_t = x_t - x_{t-1}$ and $\Delta \hat{x}_t$ are first-order differences of the ground-truth and reconstructed signals; $\mathcal{M}$ is the set of masked time steps; $\mathcal{E} = \{\, t \in \mathcal{M} \mid \text{sign}(\Delta x_t) \neq \text{sign}(\Delta x_{t+1}) \}$ marks sign-change (peak/valley) positions within the mask; and $\lambda_{\text{sign}}$ weights the extremum term ($\lambda_{\text{sign}} = 1$ by default). The first term aligns local slopes, while the second preserves peak and valley amplitudes, jointly yielding sharper and more faithful reconstructions.

**Physics-informed energy loss (PINN).** To improve generalisation and ensure that the reconstructed waveform remains physically plausible, we add a physics-informed regulariser derived from the displacement–power relationship of an IMU's proof mass. Let $x_t$ denote the reconstructed angular-rate (or acceleration) sequence inside the masked region $\mathcal{M}$. We compute the first and second discrete derivatives $\Delta x_t = x_t - x_{t-1}$, $\Delta^2 x_t = x_{t+1} - 2x_t + x_{t-1}$, and define the instantaneous *specific power* $e_t = (\Delta^2 x_{t-1} + \Delta^2 x_t)\,\Delta x_t$. Averaging over the mask gives the normalised energy

$$\bar{E} = \frac{1}{|\mathcal{M}|} \sum_{t \in \mathcal{M}} e_t, \qquad E_{\text{norm}} = \sigma(\bar{E}) \tag{3}$$

where $\sigma(\cdot)$ is the sigmoid. Extremely low or high power violates the mass–spring dynamics implicit in most MEMS sensors, so we penalise both extremes with a barrier term

$$\mathcal{L}_{\text{pinn}} = -\log\big(E_{\text{norm}}\big) - \kappa \log\big(1 - E_{\text{norm}}\big) \tag{4}$$

where $\kappa$ balances the two sides ($\kappa = 1$ in all experiments). This loss drives the reconstructed segment toward a moderate energy level, complementing the $L_2$ and correlation objectives.

## 3.2 Denoise expert

**Dual-branch complementary masking.** The Denoise Expert employs a **dual-branch MAE** whose two branches share all encoder and decoder weights[27, 28]. For each length-$L$ segment we construct two fixed 50 % masks in a cross pattern so that every even patch index is visible to branch A and masked for branch B, and vice-versa. Formally, $\mathcal{M}_A \cup \mathcal{M}_B = \{1, \dots, L\}$ and $\mathcal{M}_A \cap \mathcal{M}_B = \varnothing$, guaranteeing complementarity. Each branch receives the same noisy input but reconstructs only its own masked positions, which prevents information leakage while ensuring that no salient sample is ever hidden from both branches. After patch embedding the two masked sequences are processed by the shared encoder, padded with mask tokens, and decoded. The partial reconstructions $y_A$ and $y_B$ are fused as $y_{\text{final}} = y_A \cdot \mathcal{M}_{\mathbf{A}} + y_B \cdot \mathcal{M}_{\mathbf{B}}$, yielding a full-length denoised signal. Weight sharing regularises the model and promotes the extraction of universal features, enabling more effective suppression of high-frequency random noise.

**FFT-guided training augmentation.** Inspired by noise-injection strategies proven effective in speech enhancement[29, 30], we introduce an FFT-guided noise-injection scheme that synthesises spectrally matched corruption to create realistic training pairs. We create realistic pairs on-the-fly by injecting weak but genuine motion snippets, guided by the noise power spectrum: (1) Noise–floor estimation: For each raw noise segment we compute its FFT and obtain the power-spectral density (PSD). The median PSD value serves as the local noise floor $P_{\text{noise}}$. (2) Weak-signal injection: We randomly sample a short motion clip $s(t)$ from a separate repository of real IMU recordings (e.g., walking, hand-held rotations). The clip is amplitude-scaled to $\alpha s(t)$ with $\alpha = \beta \sqrt{P_{\text{noise}}} / \max_t |s(t)|$, where $\beta$ is a constant. The scaled clip is then added to the raw noise, yielding $x_{\text{mix}}(t) = x_{\text{noise}}(t) + \alpha s(t)$. (3) Additional corruption: After analysing the PSD, we synthesise spectrally matched noise (targeting the frequency bands that dominate QN, ARW, and BI) and add it to the mixture, producing a heavier corruption that forces the model to learn a true denoising mapping rather than smoothing $x_{\text{mix}} \leftarrow x_{\text{mix}} + x'_{\text{noise}}(t)$. (4) Training target: The mixture $x_{\text{mix}}$ is fed to the dual-branch MAE, while the reference signal is defined as $x_{\text{clean}} = x_{\text{noise}}(t) + \alpha s(t)$, without the extra corruption. This forces the network to suppress the added noise yet retain the weak real motion. This FFT-guided augmentation supplies a realistic, controllable SNR and teaches the model to enhance subtle motion cues rather than over-smooth them.

The rationale for using synthesized noise stems from established knowledge of MEMS gyroscope error sources, which are well-characterized by IEEE standards and Allan Variance analysis[31]. As any real sensor's noise profile is a composite of these known types, our FFT-guided approach is designed to be physically realistic. We first analyze the PSD of real static sensor data to identify the bands corresponding to Quantization Noise (QN), Angle Random Walk (ARW), and Bias Instability (BI). Then, we synthesize noise with a matched PSD, ensuring the key characteristics are replicated. This data-driven weak prior provides a significant advantage over injecting generic Gaussian noise or requiring expensive, perfectly-aligned ground-truth data from high-precision sensors.

## 4  Datasets & Benchmark

This section first details the datasets used for training and evaluation, and then describes **IMU Signal Enhancement Benchmark (ISEBench)**, the unified benchmark we release for fair and comprehensive assessment of IMU signal enhancement methods.

### 4.1  Datasets

We conduct all experiments on three publicly available dataset. **GyroPeak-100** (released with this paper) is a 100 Hz collection captured from the iPhone 14 on-board IMU with ground-truth peak annotations and serves as the sole source for training and evaluating the over-range reconstruction network. For the denoising task we adopt the Visual-Inertial dataset [17] and the Autonomous Platform Inertial dataset [32], both down-sampled to 100 Hz for consistency. Together, these datasets cover a broad spectrum of motion dynamics, providing a balanced and comprehensive testbed for the

proposed signal-enhancement pipeline. We follow an 80 / 20 split of each dataset for training and testing, respectively, and all experiments are executed on a single NVIDIA RTX-4060 GPU.

## 4.2 ISEBench: IMU Signal Enhancement Benchmark

To systematically and objectively evaluate the performance of the proposed inertial signal enhancement methods, we introduce and design a comprehensive testbench named **IMU Signal Enhancement Benchmark (ISEBench)**. The **ISEBench** is specifically tailored for inertial measurement unit (IMU) signal enhancement, providing a unified evaluation framework that covers multiple practical scenarios. Different from prior works that typically rely on isolated or single evaluation metrics, **ISEBench** incorporates a structured metric set categorized into three distinct aspects to thoroughly quantify enhancement performance.

**Evaluation Metrics: 1. Over-range Reconstruction Metrics:** Over-range reconstruction assesses the model's ability to recover over-range peaks that are lost when the raw signal is clipped at a dynamic-range threshold $\tau$ before being fed to the network. During evaluation we supply the model with the clipped input $x_{\text{clip}} = \text{clip}(x, \pm\tau)$ while using the unclipped signal $x$ as ground truth, and we compute all metrics only on those samples for which $|x| > \tau$. Concretely, we report **Peak Signal-to-Noise Ratio (PSNR)**, which measures the reconstruction quality of the clipped portions;

$$MSE = \frac{1}{N}\sum_{t=1}^{N}(x_t - \hat{x}_t)^2, \qquad PSNR = 10\log_{10}\left(\frac{(|\overline{\text{Peak}_{\max}}| - |\tau|)^2}{MSE}\right). \tag{5}$$

**Correlation (Corr)**, the Pearson linear correlation[33] between the reconstructed and ground-truth waveforms over the same peak regions; and **Peak Mean-Squared Error (PMSE)**, which provides a point-wise accuracy measure at the detected peak locations. Let $\mathcal{P}$ be the index set of local peaks;

$$\text{PMSE} = \frac{1}{|\mathcal{P}|}\sum_{t\in\mathcal{P}}(y_t - \hat{y}_t)^2. \tag{6}$$

**2. Weak Signal Enhancement Metric:** This metric assesses the capability of the proposed approach in extracting and enhancing low-amplitude signals: **Signal-to-Noise Ratio (SNR)**[34] evaluates weak signal recovery effectiveness. **3. Static Noise Performance Metrics:** To characterize sensor performance in static (non-moving) conditions, we adopt standard inertial measurement unit performance metrics defined by Allan variance[31], including: **Angle Random Walk (ARW)**, **Quantization Noise (QN)** and **Bias Instability (BI)**.These metrics are computed following standard Allan variance analysis methodology described comprehensively in prior studies. The formulas for the above seven metrics are given in the Appendix A.

Together, these metrics constitute **ISEBench**, a unified and transparent yardstick for inertial-signal–enhancement research. In the following experiments we leverage **ISEBench** to benchmark our model against state-of-the-art baselines, highlighting its effectiveness.

# 5 Experiments

## 5.1 Comparison with Previous Results

For the quantitative comparison, we pit **MoE-Gyro** against nine carefully reproduced baselines drawn from three methodological families: classic model-driven signal processors (EMD, Savitzky–Golay filtering, and the Matlab over-range signal reconstruction function); fully supervised deep networks (CNN, kNN, LSTM–GRU and HEROS_GAN); and the self-supervised model, IMUDB. To ensure fairness, all supervised baselines are retrained on matched data: clipped and full pairs from our Peak database for over-range reconstruction and clean/noisy pairs from the *Autonomous Platform Inertial* dataset for denoising. Each method is executed exactly as specified in its original paper, using the authors' code when available or a validated re-implementation otherwise. The resulting performance in **ISEBench** is summarized in Table 1. **MoE-Gyro** attains the best average rank across all metrics.

We analyze the performance of different methods separately for over-range reconstruction and denoising tasks. Regarding over-range reconstruction, Table 1 shows that classical model-driven methods indeed raise PSNR and Corr, but their P_MSE remains large, revealing limited accuracy at the peak locations. Because these methods extrapolate by incrementally fitting the visible portion

Table 1: Performance comparison on **ISEBench**. The best result is **boldfaced** and the second best is underlined. For clarity, each P_MSE entry is written as $P\_MSE_i/P\_MSE_{RAW}$, and all Allan-variance metrics are reported as the percentage reduction relative to the raw signal (the omitted values are provided in Appendix C).

| Model\Metric | Peak Rec. ($\tau = 450°/s$) | | | Weak Sig. | Allan Variance | | | AVG.rank |
| | PSNR ↑ | P_MSE ↓ | Corr ↑ | SNR ↑ | QN ↓ | ARW ↓ | BI ↓ | |
| | dB | - | - | dB | (°/s) | (°/√h) | (°/h) | |
|---|---|---|---|---|---|---|---|---|
| RAW | 2.67 | 1 | - | 10.18 | 0 | 0 | 0 | - |
| Matlab 2023[19] | 6.03 | 0.515 | 0.86 | 10.02 | -25.8% | -3.1% | +5.4% | 7.0 |
| EMD[20] | 5.44 | 0.655 | 0.77 | 13.85 | -91.1% | -85.9% | -96.8% | 5.3 |
| SG_filter[21] | 4.35 | 0.767 | 0.79 | 12.03 | -85.0% | -86.3% | -90.0% | 6.6 |
| CNN[13] | 5.76 | 0.621 | 0.85 | 14.3 | -62.5% | -35.9% | -79.4% | 6.1 |
| LSTM_GRU[14] | 5.95 | 0.495 | 0.87 | 19.23 | -80.8% | -85.0% | -93.1% | 4.3 |
| KNN[32] | 3.72 | 0.752 | 0.67 | 12.54 | -85.7% | -34.3% | -47.5% | 7.7 |
| HEROS_GAN[15] | 7.7 | 0.354 | 0.89 | 16.86 | -92.8% | -51.6% | -58.3% | 3.6 |
| IMUDB[17] | 6.59 | 0.442 | 0.82 | 17.76 | -85.8% | -87.8% | -93.7% | 3.4 |
| **MoE-Gyro** | **8.29** | **0.325** | **0.92** | **24.19** | **-98.0%** | **-94.1%** | **-98.4%** | **1** |

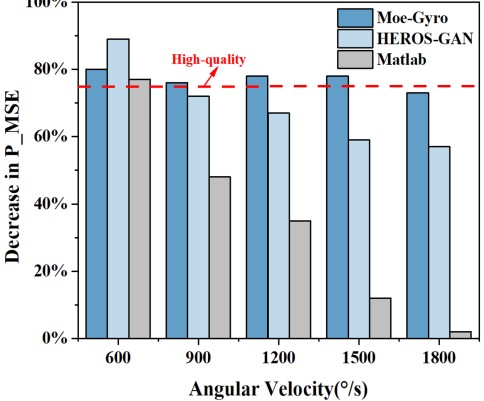

Figure 2: Comparison of reconstruction P_MSE. We compare **MoE-Gyro** with two representative baselines,a drop of more than 75 % (dashed reference) marks high-quality recovery.

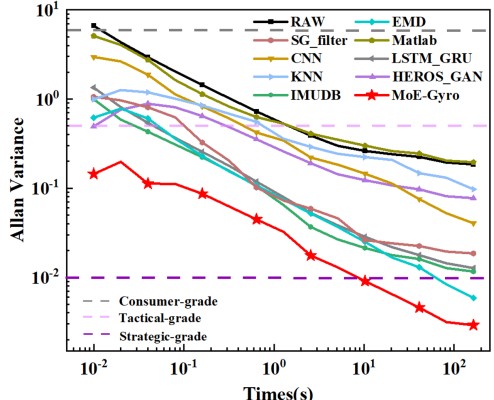

Figure 3: Allan-variance comparison. The red curve corresponds to **MoE-Gyro** and shows the best Allan-variance performance—raising the device from consumer to nearly strategic grade.

of the waveform. Meanwhile, we compare **MoE-Gyro** with the best data-driven baseline (HEROS-GAN) and the best model-driven baseline (Matlab2023) by plotting the relative P_MSE reduction at different angular velocity thresholds (Fig. 2). At $1500°/s$ our method still achieves high-quality peak reconstruction, whereas HEROS-GAN deteriorates noticeably beyond $900°/s$; the model-driven filter ceases to reconstruct peaks effectively once the angular velocity exceeds $600°/s$. These results further substantiate the limitations discussed above. **MoE-Gyro** outperforms all baselines thanks to (i) an adaptive MAE mask that keeps peak-relevant patches, (ii) Gaussian-Decay Attention that concentrates decoding on the clipped region, and (iii) a carefully engineered loss that restores local dynamics (see ablations).

In terms of denoising performance, classical filters lower Allan-variance terms but also erase faint motion, so SNR barely improves. Supervised deep-learning baselines, trained as single multitask networks, likewise sacrifice SNR because the much larger reconstruction loss dominates optimisation. By routing segments to a dedicated Denoise Expert and training it with FFT-guided noise augmentation, **MoE-Gyro** delivers the highest SNR while further reducing QN, ARW, and BI; Allan-variance curves in Fig.3 visualise the gain.

## 5.2 Real-world Experiment

To intuitively demonstrate the practical capability of our framework in handling severe over-range conditions, we randomly select a representative segment from the test set for visualization (Fig. 4).

Table 2: Cross-Device Generalization results

| Device | PSNR↑ | ARW Reduction↑ |
|---|---|---|
| iPhone 14 | **8.29** /7.70 | **94.1%**/51.6% |
| Xiaomi 14 | **7.95** /7.51 | **86.5%**/48.3% |
| Huawei P70 | **8.28** /7.41 | **88.9%**/45.7% |

Table 3: Cross-Task Generalization results

| Task Type | PSNR↑ |
|---|---|
| Running Swing | **8.33** / 7.04 |
| Jumping | **7.70** / 7.73 |
| Wrist Twist | **8.18** / 7.56 |

The segment includes measurements from a low-range IMU (IM900, ±450°/s), the corresponding ground truth, and the enhanced output from our **MoE-Gyro**. At the highlighted peak, the true angular velocity reaches -1731.8°/s, significantly exceeding the measurement limit of the IM900 sensor. Despite this substantial clipping, **MoE-Gyro** effectively reconstructs the peak to -1453.7°/s, accurately capturing key signal dynamics beyond the nominal range. By contrast, the best competing method we tested lifts the same peak only to -1287 °/s and exhibits pronounced waveform distortion around the apex, detailed traces are provided in Appendix C. This capability suggests substantial potential for expanding the practical utility of low-cost, limited-range inertial sensors.

## 5.3 Generalization and Robustness

To be practical, a signal enhancement framework must generalize across different hardware and dynamic conditions. We evaluated **MoE-Gyro'** zero-shot generalization capability and its performance on distinct motion types. (**MoE-Gyro's** results are **boldfaced**.)

**Cross-Device Generalization.** Our model trained on data from the iPhone 14, was tested directly on new data collected from a Huawei P70 and a Xiaomi 14 without any fine-tuning. For new devices, signals only require resampling to 100Hz input rate and unit conversion to rad/s. As shown in the Table 2, **MoE-Gyro** demonstrates strong zero-shot performance, with minimal degradation in over-range reconstruction and robust noise reduction compared to the HEROS-GAN baseline. The slight decline in denoising is expected, as baseline noise characteristics differ between sensors.

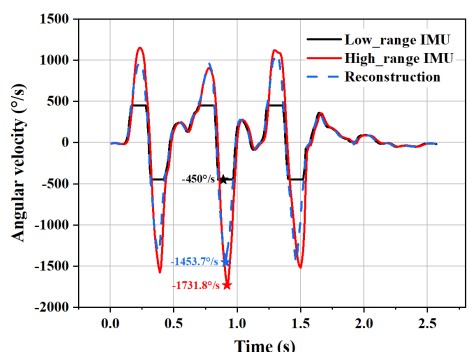

Figure 4: Range-extension visualisation. At an actual angular rate of –1731.8°/s, our method reconstructs the signal clipped at the 450°/s sensor limit to –1453.7°/s.

**Cross-Task Generalization.** We further analyzed over-range reconstruction performance across different motion dynamics present in our dataset: periodic swings (e.g., running), high-amplitude impacts (e.g., jumping), and high-frequency twists. As shown in Table 3, results confirm that **MoE-Gyro** maintains stable, high-level performance across these distinct tasks, indicating that it learns general principles of signal dynamics rather than overfitting to specific motion patterns.

## 5.4 Efficiency and Real-Time Performance

For real-world applications like robotics and UAVs, computational efficiency is critical. We assessed **MoE-Gyro's** performance from the perspectives of PC-based real-time processing and embedded deployment feasibility.

The full **MoE-Gyro** model processes a 2.56-second data segment in 117 ms on an NVIDIA

Table 4: Performance and Efficiency comparison

| Model | PSNR↑ | SNR↑ | Inf Time ↓ |
|---|---|---|---|
| Ours (Full) | 8.29 | 24.19 | 117ms |
| **Ours (Comp.)** | 7.89 | 21.60 | 41ms |
| Heros_GAN | 7.70 | 16.86 | 39ms |

RTX 4060 GPU. This allows for an overlapping sliding-window approach with a 0.5-second step size, achieving a 2Hz update rate suitable for many mid-level state estimation tasks. The framework acts as a Navigation Co-processor, providing high-quality motion updates to supplement a system's main navigation filter, rather than operating in the low-level, high-frequency control loop.

| Table 5: MoE ablation results. | | | |
|---|---|---|---|
| Model | PSNR↑ | SNR↑ | Mem↓ |
| ORE+DE | 8.19 | 24.58 | 129 MB |
| SingleNet | 7.23 | 12.9 | 64.7 MB |
| **MoE-Gyro** | **8.29** | **24.19** | **71.3 MB** |

| Table 6: Impact of GD-Attn Placement | | | |
|---|---|---|---|
| Setting | PSNR↑ | P_MSE↓ | Corr↑ |
| No GD-Attn | 8.08 | 0.345 | 0.87 |
| Encoder-only | 8.03 | 0.345 | 0.88 |
| Enc.+Dec. | 8.27 | 0.335 | 0.90 |
| **Decoder-only** | **8.29** | **0.324** | **0.92** |

To evaluate embedded deployment potential, we compressed the model via pruning, reducing its parameter count by 12x to 1.85M, comparable to lightweight architectures like MobileNetV2. The compressed model significantly improves inference speed with only a minor drop in performance(Tab.4), making it a viable candidate for deployment on edge AI platforms like NVIDIA Jetson.

## 5.5 Ablation Studies

In this section, we conduct ablation experiments on the components of our method. By systematically enabling or disabling each component, or substituting it with simpler counterparts, we quantify how much performance each element contributes to the final system.

**Ablation on the MoE.** To assess the impact of the MoE architecture itself, we compare alternative expert-invocation schemes with a single multi-task network of equal size. Table 5 shows that the full MoE-Gyro achieves virtually identical enhancement to calling both the ORE and DE, indicating that the router and concatenation do not degrade either task. A single multi-task network of identical size trails behind on both metrics, confirming the advantage of explicit task decoupling. Since most of the time the MoE architecture calls only a single expert to process a segment, **MoE-Gyro** requires roughly half GPU memory consumed when both experts are run unconditionally, while matching their combined quality, demonstrating a clear efficiency gain.

**Ablation on the Over-range Reconstruction Expert.** We systematically evaluate the individual contributions of GD-Attn, the correlation loss, and the PINN regularizer in our Over-range Reconstruction Expert. First, we investigate the optimal placement of the GD-Attn module (Tab. 6). Results show decoder-only integration of GD-Attn achieves the largest improvement. This demonstrates GD-Attn's critical role in refining latent representations specifically at the decoding stage, effectively focusing reconstruction capacity on clipped regions.

| Table 7: PINN vs. Second-Order Smoothness. | | | |
|---|---|---|---|
| Loss | PSNR↑ | P_MSE↓ | Corr↑ |
| Smoothness | 8.02 | 0.354 | 0.88 |
| **PINN** ($\kappa=1$) | **8.29** | **0.324** | **0.92** |

Second, we compare our proposed physics-informed energy loss (PINN) against a conventional second-order smoothness prior (Tab. 7)[35, 36]. When evaluated on unseen data, our PINN consistently outperforms the smoothness regularizer, improving PSNR by 0.27 dB and reducing P_MSE by 8%, highlighting the strong generalization ability provided by the physics-based constraint.

Finally, we comprehensively explore all combinations to quantify their cumulative and complementary effects (Tab. 8). Individually,

| Table 8: Component ablation results. | | | |
|---|---|---|---|
| Model | PSNR↑ | P_MSE↓ | Corr↑ |
| No Components | 7.68 | 0.369 | 0.88 |
| GD-Attn | 7.96 | 0.364 | 0.90 |
| Corr | 7.83 | 0.354 | 0.91 |
| PINN | 7.95 | 0.345 | 0.90 |
| GD+Corr | 8.21 | 0.350 | 0.91 |
| GD+PINN | 8.19 | 0.339 | 0.91 |
| Corr+PINN | 8.08 | 0.340 | 0.91 |
| **All** | **8.29** | **0.324** | **0.92** |

GD-Attn significantly improves peak restoration, correlation loss notably enhances waveform fidelity, and PINN markedly stabilizes signal reconstruction. Jointly, these components interact positively, resulting in the best overall trade-off in reconstruction quality and stability.

**Ablation on the Denoise Expert.** We isolate and analyze three key design choices within the denoising expert: mask strategy, weight sharing, and augmentation strategy. First, we optimize the complementary mask ratio from 0% to 50% (Fig. 5a), finding that a dual-branch architecture with

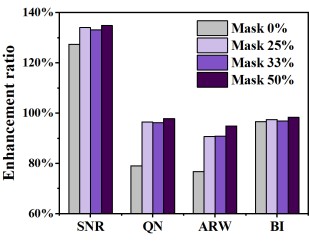 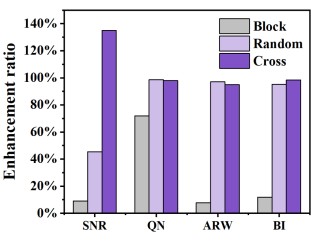 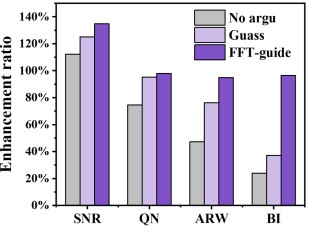

(a) Mask-ratio comparison      (b) Mask-pattern comparison      (c) FFT augmentation ablation

Figure 5: Ablation studies for the Denoise Expert. (a) Mask ratio: a 50 % mask yields the strongest denoising effect; (b) Mask pattern: the cross mask outperforms block and random patterns; (c) FFT augmentation: FFT-guided noise injection gives the largest performance gain.

a 50% mask ratio consistently yields superior noise suppression performance. Next, holding this ratio fixed, we compare different mask patterns (Fig. 5b). A block mask removes large contiguous regions and thus underperforms significantly, while random masking shows improvements on Allan metrics but still exhibits suboptimal SNR due to residual noise leakage. Our deterministic cross mask achieves the best balance, matching random masking on Allan metrics while outperforming in SNR, attributed to complete temporal coverage and minimal information gaps.

We then examine the effectiveness of parameter sharing between branches (Tab. 9). While independent branches double model capacity, full encoder-decoder weight sharing reduces parameters by 50% and GPU memory usage by 45% with negligible impact on denoising quality. Partial sharing variants either reduce efficiency gains or degrade signal qual-

Table 9: Weight-Sharing study

| Weight sharing | SNR↑ | GPU mem↓ | Params↓ |
|---|---|---|---|
| No share | 24.51 | 116.8 | 27.8 |
| E share | 24.35 | 76.2 | 17.15 |
| D share | 24.27 | 105.0 | 24.64 |
| **E+D share** | **24.19** | **64.4** | **13.9** |

ity. Thus, full weight sharing represents the optimal complexity-performance trade-off and is adopted.

Finally, we evaluates the impact of our FFT-guided augmentation strategy(Fig. 5c). Compared to a baseline without augmentation, Gaussian noise injection modestly improves SNR but leaves Allan metrics largely unaffected. Our FFT-guided augmentation introducing spectrally matched synthetic noise, achieves comprehensive gains across all metrics. These results validate the superiority of using realistic spectral characteristics to train a more robust denoiser.

## 6 Conclusion

In this work, we introduced **MoE-Gyro**, a novel self-supervised Mixture-of-Experts framework tailored to simultaneously address the long-standing trade-off between measurement range and noise performance in MEMS gyroscopes. **MoE-Gyro** leverages masked autoencoder (MAE) architectures to independently optimize two dedicated experts: an Over-Range Reconstruction Expert (ORE), enhanced by Gaussian-Decay Attention and physics-informed constraints for accurate reconstruction of saturated signals; and a Denoise Expert (DE), utilizing complementary masking and FFT-guided augmentation to significantly reduce noise without requiring labeled data. Additionally, we introduced **ISEBench**, the first open-source evaluation benchmark designed for fair and comprehensive assessment of IMU signal-enhancement methods. Experiments on **ISEBench** demonstrated that **MoE-Gyro** extends the measurable range from ±450 °/s to ±1500 °/s and reduces Bias Instability by 98.4%, significantly surpassing existing baselines. Despite these advancements, the relatively large size of our proposed architecture may pose challenges for resource-constrained embedded deployments; future work could thus explore model compression techniques to enable more efficient deployment. Our study opens a promising path to deep learning-based performance upgrades for MEMS inertial sensors and establishes a solid baseline for future research in inertial signal enhancement.

## Acknowledgements

The authors would like to express their sincere gratitude to Ph.D. candidate Yongchao Li and Master's student Tao Chen from Southeast University for their invaluable support and thoughtful guidance. This work was supported by the School of Integrated Circuits, Southeast University.

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

# A    Metric Definitions in ISEBench

For completeness we list the analytical forms of the seven evaluation metrics used throughout ISEBench. Let $y = \{y_t\}_{t=1}^N$ be the ground-truth sequence, $\hat{y} = \{\hat{y}_t\}_{t=1}^N$ the reconstructed (or denoised) sequence, and $\mathcal{P} \subseteq \{1, \ldots, N\}$ the set of local peaks or valleys indices.

## A.1    Over-range reconstruction metrics

**Peak-averaged PSNR.**    For each test segment $s$ we first locate its maximum absolute in-range value $\text{Peak}_{\max}^{(s)} = \max_{t \in s} |y_t|$ and then compute the dataset-level mean peak

$$\overline{\text{Peak}}_{\max} = \frac{1}{S} \sum_{s=1}^{S} \text{Peak}_{\max}^{(s)}. \tag{7}$$

Let Clip denote the sensor's full-scale range (e.g. $\pm 450°/\text{s}$). The peak-averaged PSNR is defined as

$$\text{PSNR} = 10 \log_{10} \left( \frac{\left(\overline{\text{Peak}}_{\max} - \text{Clip}\right)^2}{\frac{1}{N} \sum_{t=1}^{N} (y_t - \hat{y}_t)^2} \right). \tag{8}$$

Here the numerator reflects the *average recoverable headroom* between the sensor's clip level and the typical (mean) peak amplitude, yielding a scale that is consistent across all segments.

**Correlation (Corr).**    Corr measures the linear similarity between the reconstructed signal $\hat{y}$ and the ground truth $y$, serving as an indicator of reconstruction linearity. It is defined as:

$$\text{Corr} = \frac{\sum_{t=1}^{N} (y_t - \bar{y})(\hat{y}_t - \bar{\hat{y}})}{\sqrt{\sum_{t=1}^{N}(y_t - \bar{y})^2} \sqrt{\sum_{t=1}^{N}(\hat{y}_t - \bar{\hat{y}})^2}} \tag{9}$$

where $\bar{y} = \frac{1}{N} \sum_{t=1}^{N} y_t$ and $\bar{\hat{y}} = \frac{1}{N} \sum_{t=1}^{N} \hat{y}_t$ are the mean values of the true and reconstructed sequences, respectively, over $N$ samples.

**Peak Mean Squared Error (P_MSE).**    P_MSE directly measures the average reconstruction error at the clipped-peak locations, quantifying the fidelity of peak recovery. Let $\mathcal{P}$ be the set of time indices where the true signal exceeds the sensor's range (i.e. the clipped samples). Then

$$\text{P\_MSE} = \frac{1}{|\mathcal{P}|} \sum_{t \in \mathcal{P}} (y_t - \hat{y}_t)^2 \tag{10}$$

where $y_t$ is the ground truth and $\hat{y}_t$ the reconstructed value at sample $t$.

## A.2    Weak Signal Enhancement Metric

**Signal-to-Noise Ratio (SNR).**    SNR quantifies the relative power of the desired signal versus background noise. Let $\{s_t\}_{t=1}^N$ be the segment of interest and $\{n_t\}_{t=1}^N$ the corresponding noise-only sequence. We define

$$P_{\text{signal}} = \frac{1}{N} \sum_{t=1}^{N} s_t^2, \qquad P_{\text{noise}} = \frac{1}{N} \sum_{t=1}^{N} n_t^2. \tag{11}$$

Then

$$\text{SNR} = 10 \log_{10}\left(\frac{P_{\text{signal}}}{P_{\text{noise}}}\right). \tag{12}$$

### A.3 Weak Signal Enhancement Metric

**Allan Variance Computation.** Given a sequence of angular-rate measurements $\{x_k\}$ sampled at interval $T_0$, we first form non-overlapping averages over clusters of length $\tau = m\,T_0$:

$$\bar{x}_i(\tau) = \frac{1}{m} \sum_{k=(i-1)m+1}^{im} x_k, \quad i = 1, \ldots, M, \tag{13}$$

where $M = \lfloor N/m \rfloor$. The Allan variance at cluster time $\tau$ is then

$$\sigma_y^2(\tau) = \frac{1}{2(M-1)} \sum_{i=1}^{M-1} \left( \bar{x}_{i+1}(\tau) - \bar{x}_i(\tau) \right)^2, \tag{14}$$

and the Allan deviation is $\sigma_y(\tau) = \sqrt{\sigma_y^2(\tau)}$.

From $\sigma_y(\tau)$ we derive three standard performance parameters:

$$\text{QN} = \frac{\sigma_{-1}(1)}{\sqrt{3}} \quad \left( \text{slope} -1 \text{ at } \tau = 1 \right) \tag{15}$$

$$\text{ARW} = \sigma_{-\frac{1}{2}}(1) \quad \left( \text{slope} -\tfrac{1}{2} \text{ at } \tau = 1 \right) \tag{16}$$

$$\text{BI} = \sigma_{y,\min} \sqrt{\frac{2 \ln 2}{\pi}} \tag{17}$$

Here, $\sigma_{-1}(1)$ denotes the value of $\sigma_y(\tau)$ at $\tau = 1$ extrapolated along the slope –1 region, and $\sigma_{-\frac{1}{2}}(1)$ denotes the analogous intercept for the slope -1/2 region. Within the testbench, QN and ARW are extracted from the log–log slopes of the raw signal's Allan deviation curve and $\sigma_{y,\min}$ is the minimum deviation used to compute bias instability.

## B  Derivation of the Physics-Informed Energy Loss

**Mechanical background.** For a MEMS proof mass of unit mass ($m=1$) moving along one axis, the instantaneous mechanical energy is $\mathcal{E}(t) = \frac{1}{2}v^2(t) + \frac{1}{2}kx^2(t)$, where $x(t)$ and $v(t) = \dot{x}(t)$ are displacement and velocity, and $k$ is the effective spring constant. The specific power(time rate of change of energy per unit mass) is

$$\mathcal{P}(t) = \frac{\mathrm{d}\mathcal{E}}{\mathrm{d}t} = a(t)\,v(t), \qquad a(t) = \ddot{x}(t). \tag{18}$$

**Discrete approximation.** Our network reconstructs a discrete angular-rate (or acceleration) sequence $\{x_t\}_{t\in\mathbb{Z}}$ with unit sample period ($\Delta t=1$). We approximate the first and second derivatives by

$$\Delta x_t = x_t - x_{t-1}, \qquad \Delta^2 x_t = x_{t+1} - 2x_t + x_{t-1}. \tag{19}$$

Substituting (16) into (17) and centring the acceleration term yields a discrete specific power

$$e_t = \underbrace{\left( \tfrac{1}{2}\Delta^2 x_{t-1} + \tfrac{1}{2}\Delta^2 x_t \right)}_{\text{acc. at } t} \Delta x_t. \tag{20}$$

**Mask-averaged normalised energy.** Given the set $\mathcal{M}$ of masked (to-be-reconstructed) indices, we take the mean specific power

$$\bar{E} = \frac{1}{|\mathcal{M}|} \sum_{t\in\mathcal{M}} e_t, \tag{21}$$

and pass it through a sigmoid $E_{\text{norm}} = \sigma(\bar{E}) \in (0,1)$ to bound the value and allow symmetrical penalties on both extremes.

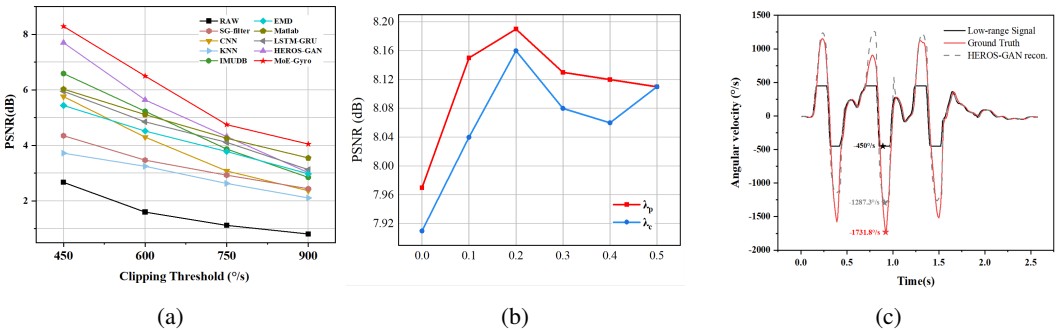

Figure 6: Additional Experiments. (a) Clipping-Threshold Analysis. **MoE-Gyro** maintains the best performance across all tested saturation, model-driven methods improve in relative ranking as the threshold rises. (b) Sensitivity analysis of $\lambda$ in loss; (c) Reconstruction by HEROS-GAN.

**Barrier formulation.** Very small $E_{\text{norm}}$ corresponds to an over-damped (excessively smooth) reconstruction, whereas values near one indicate unphysical high-frequency ringing. We therefore introduce the barrier

$$\mathcal{L}_{\text{pinn}} = -\log\big(E_{\text{norm}}\big) - \kappa \, \log\big(1 - E_{\text{norm}}\big). \tag{22}$$

where $\kappa$ balances the penalty on the high-energy side ($\kappa{=}1$ in our experiments). Minimising (20) drives $E_{\text{norm}}$ toward a moderate value, enforcing a physically plausible energy level while remaining fully differentiable.

## C  Supplementary Experiments and Data

**Analysis on clipping threshold.** In the main text we report results for a clipping threshold of $\pm450°/\text{s}$. To evaluate robustness under more severe saturation, we additionally clip all test signals at $\pm600°/\text{s}$, $\pm750°/\text{s}$ and $\pm900°/\text{s}$ and repeat the reconstruction experiments. For each clipping range, the corrupted signals are processed by our Over-Range Expert and the PSNR is computed.

As depicted in Figure 6a, increasing the clipping threshold naturally reduces the average recoverable headroom $\overline{\text{Peak}}_{\text{max}} - \text{Clip}$, leading to lower absolute PSNR values. Crucially, however, our Over-Range Expert still delivers substantial PSNR gains over the raw clipped signals at every tested threshold, demonstrating its robustness across sensors with varying measurement ranges.

**Loss-weight sensitivity.** The total reconstruction loss is $\mathcal{L} = L_2 + \lambda_c \, L_{\text{corr}} + \lambda_p \, L_{\text{pinn}}$. We fix $\lambda_p = 0.5$ and vary $\lambda_c$ from 0.1 to 0.5, then fix $\lambda_c = 0.5$ and vary $\lambda_p$ over the same range. As shown in Figure 5(b), the highest PSNR is achieved at $(\lambda_c, \lambda_p) = (0.5, 0.2)$. We also evaluated the symmetric setting ($\lambda_c = \lambda_p = 0.2$), which underperformed the asymmetric combination. Therefore, we adopt $\lambda_{L2} = 1$, $\lambda_c = 0.5$, and $\lambda_p = 0.2$ for all over-range reconstruction experiments.

**Real-World Over-Range Reconstruction with HEROS-GAN.** To benchmark against the current state of the art, we applied HEROS-GAN[15] to the same real-world test segment (Fig.6c). While HEROS-GAN succeeds at reconstructing the overall waveform shape, it systematically underestimates extreme values—recovering the highlighted peak to only –1287.3°/s—and introduces spurious oscillations. This behavior underscores its limited capacity for precise peak recovery under severe clipping.

**Supplementary Data for Table 1.** For completeness, all numerical entries omitted from Table 1 are reported here in Table 10.

Table 10: Full benchmark results corresponding to Table 1.

| Model\Metric | Peak Rec. ($\tau = 450°/s$) | Allan Variance | | |
| | P_MSE ↓ | QN ↓ | ARW ↓ | BI ↓ |
| | $(°/s)^2$ | $(°/s)$ | $(°/\sqrt{h})$ | $(°/h)$ |
|---|---|---|---|---|
| RAW | 181456 | 0.0004 | 0.32 | 10.03 |
| Matlab 2023 | 93371 | 0.0002956 | 0.31 | +5.4% |
| EMD | 118916 | 0.0000357 | 0.045 | 0.32 |
| SG_filter | 139175 | 0.00006 | 0.044 | 1.0 |
| CNN | 112750 | 0.00015 | 0.205 | 2.07 |
| LSTM_GRU | 89847 | 0.0000767 | 0.048 | 0.69 |
| KNN | 136533 | 0.0000573 | 0.21 | 5.27 |
| HEROS_GAN | 64302 | 0.0000287 | 0.155 | 4.18 |
| IMUDB | 80158 | 0.0000567 | 0.039 | 0.63 |
| **MoE-Gyro** | **59017** | **0.000008** | **0.019** | **0.157** |

# D    More Discussions and Future Directions

## D.1    Synergistic Effects of Over-Range Signals and Noise

Noise in MEMS gyroscopes can be categorized as either internal device noise (thermo-mechanical, electronic) or undesired external inputs (e.g., mechanical shocks). Internal noise defines the sensor's baseline noise floor and its amplitude is insufficient to cause saturation. In contrast, high-amplitude external inputs are legitimate physical signals that can cause saturation, resulting in a complex mixed signal containing both clipped peaks and potential post-shock ringing.

Our MoE architecture is explicitly designed to handle both scenarios. The Denoise Expert (DE) filters low-amplitude internal noise, while the Over-Range Reconstruction Expert (ORE) recovers the waveform of saturated segments caused by external inputs. The gate mechanism routes segments appropriately, allowing the framework to decouple the distinct mathematical objectives of reconstruction and denoising. This design ensures the delivery of a high-quality, wide-range, and low-noise signal, forming a reliable foundation for downstream navigation and control tasks.

## D.2    Ablation Study on Gating Mechanism

Our choice of a rule-based heuristic gate was motivated by its near-zero computational overhead and predictable behavior. To explore alternatives, we trained a lightweight MLP to act as a learned gate for classifying signal segments. We compared its performance against the heuristic gate on the mixed-signal dataset. The MLP gate showed a stronger ability to identify noise, leading

Table 11: Gate Ablation

| Gate Type | PSNR↑ | SNR↑ |
|---|---|---|
| Heuristic gate | 7.92 | 22.89 |
| MLP | 7.65 | 23.91 |

to improved SNR, but was sometimes overly conservative and failed to trigger the ORE when needed, slightly reducing PSNR.(Tab.11) While the heuristic gate provided the best overall balance for our current needs, exploring more advanced neural gate architectures remains a key direction for future work.

## D.3    Downstream Impact of PINN Regularization

To provide stronger evidence for the role of the Physics-Informed Neural Network (PINN) loss as a regularizer, we conducted an ablation study assessing its impact across diverse dynamic tasks. We compared the performance of the full model (with PINN) against a version without it.

While the unconstrained model achieves a slightly higher PSNR on the simpler, periodic Running Swing task, the PINN-constrained model demonstrates far more stable and consistent performance across all tasks. Its performance does not fluctuate drastically with changes in signal dynamics. This highlights the core function of PINN: it trades a small amount of specialization on simple tasks

for significantly improved robustness and generalization on complex and varied motion patterns by enforcing physical plausibility.

Table 12: Downstream Impact of PINN

| Task Type | PSNR(with PINN)↑ | PSNR(without PINN)↑ |
|---|---|---|
| Running Swing | 8.33 | 8.48 |
| Jumping | 7.70 | 7.39 |
| Wrist Twist | 8.18 | 8.01 |

### D.4 Sensor Fusion Applications

Beyond single-sensor signal enhancement, the methodologies presented in this work, particularly the Denoise Expert (DE), hold significant promise for multi-sensor systems such as Visual-Inertial Odometry (VIO) and SLAM. The DE can serve as a powerful front-end IMU data calibrator, providing a high-quality, low-noise angular velocity and acceleration stream to the main navigation estimator.

A primary challenge in IMU-based navigation is the rapid accumulation of orientation error, especially the unobservable drift in the yaw angle, which severely degrades long-term trajectory accuracy. By effectively suppressing Angle Random Walk (ARW) and Bias Instability (BI), our framework can provide a more stable and reliable inertial input. This corrected IMU data can be pre-integrated and tightly coupled with visual features from a camera, forming a high-precision VIO system. Alternatively, for applications where computational efficiency is paramount, our method enables an IMU-dominant fusion scheme. In this configuration, the high-quality inertial data drives the state propagation, with camera observations providing periodic corrections, resulting in a robust and efficient VIO pipeline. This positions our work as a valuable preprocessing step to enhance the accuracy and robustness of downstream multi-sensor fusion tasks.

### D.5 Discussion on Iterative Denoising

One potential limitation of our current self-supervised training paradigm is that the "low-noise" weak motion signals, used as training targets are not perfectly noise-free. While they represent a significant improvement over the synthetically corrupted inputs, they inherently contain some level of residual noise from the sensor's baseline noise floor. This suggests that the model's performance ceiling is tied to the quality of the training repository.

This observation opens a promising avenue for future work: an iterative self-purification training scheme. The core idea is to leverage the currently trained MoE-Gyro model to further refine its own training data. The process would involve two stages: 1. Dataset Purification: Use the trained Denoise Expert to perform an initial denoising pass on the entire repository of weak motion signals, creating a "cleaner" set of signals. 2. Model Retraining: Retrain the Denoise Expert from scratch, using this purified dataset as a more ideal training target.

This loop could theoretically be repeated, allowing the model to progressively improve its understanding of the underlying signal characteristics by iteratively reducing the noise in its training targets. This concept of iterative refinement to enhance signal estimation shares conceptual similarities with advanced techniques in other signal processing domains, such as learning to smooth in partially known state-space models[37]. Exploring this self-improving training strategy presents a valuable direction for pushing the performance boundaries of IMU signal enhancement.

## E  Broader Impacts

This work aims to improve inertial-sensor signal quality through a self-supervised Mixture-of-Experts framework. Enhanced MEMS gyroscopes can benefit a wide spectrum of applications—from consumer electronics and robotics to autonomous navigation—by enabling higher accuracy without added hardware complexity. While better motion sensing may indirectly influence downstream systems (e.g., drones, vehicles, or defense technologies), we do not foresee any immediate, unique societal risks posed by the algorithm itself beyond those already associated with general improvements in sensor signal processing.

