# OpenReview forum: "MoE-Gyro: Self-Supervised Over-Range Reconstruction and Denoising for MEMS Gyroscopes"
_NeurIPS.cc/2025/Conference — NeurIPS 2025 poster_

### Official Review · Reviewer_Ztva · 2025-06-30

**Clarity:** 4
**Significance:** 3
**Originality:** 3
**Rating:** 5
**Confidence:** 4

**Summary:**

This paper proposes a novel self-supervised framework, MoE-Gyro, to simultaneously address over-range signal reconstruction and noise reduction in MEMS gyroscopes which commonly used in inertial navigation but limited by a trade-off between range and noise. The architecture uses a Mixture-of-Experts (MoE) approach with two specialized branches: an Over-Range Reconstruction Expert (ORE) enhanced by Gaussian-Decay Attention and physics-informed loss, and a Denoise Expert (DE) equipped with dual-branch masking and FFT-guided augmentation. A lightweight gating mechanism dynamically routes signal segments to the appropriate expert. The authors also introduce ISEBench, a new benchmark, and show that MoE-Gyro significantly outperforms existing methods.

**Questions:**

See weaknesses 1-4.

**Ethical Concerns:**

["NO or VERY MINOR ethics concerns only"]

**Final Justification:**

Accept.

**Limitations:**

See weakness 1.

The reviewer believes that a discussion of alternative MoE architectures (such as multipath or non-exclusive routing) and their implications on training dynamics, as well as the potential modifications required in other components of the design, would strengthen the paper (interests a wider audience and followup work).

**Quality:**

3

**Strengths And Weaknesses:**

Strengths

1. The paper is motivated by a practical real-world challenge, i.e., the inherent limitations of current MEMS gyroscope systems, particularly the trade-off between extending measurement range and achieving effective noise reduction.
2. Motivated by the observation that naively unifying denoising and over-range reconstruction leads to reconstruction loss dominating while the denoising loss is largely ignored, the paper proposes a unified self-supervised framework based on a Mixture-of-Experts (MoE) architecture. Both the Over-Range Reconstruction Expert (ORE) and the Denoise Expert (DE) are thoughtfully and appropriately designed for their respective tasks.
3. The paper introduces a new benchmark, ISEBench, which comprises two publicly available datasets along with a newly collected dataset by the authors. The benchmark also features a set of thoughtfully designed evaluation metrics to enable comprehensive assessment of IMU signal enhancement methods.
4. The evaluation is thorough, including both baseline comparisons and ablation studies, as well as a real-world experiment which the reviewer finds commendable and appreciates.

Weaknesses

1. While the design of the MoE architecture and routing logic is well motivated by the observation that naively combining denoising and over-range reconstruction leads to training imbalance, it remains unclear to the reviewer whether the strict either-or routing choice is the most appropriate or only viable solution. It is plausible that a signal segment could be both clipped and noisy. In fact, when a sensor experiences repeated clipping events, the resulting signal may exhibit increased noise or instability, suggesting that simultaneous application of both experts could be beneficial in some cases.
2. Although the router is lightweight, it is based on hard-coded heuristics rather than a learned gating mechanism, and it does not provide any form of confidence estimation for its expert selection.
3. The PINN constraint employed is a simplified energy-based loss that does not explicitly enforce the true physical dynamics of the system. This simplicity could be considered a strength, provided stronger empirical evidence (particularly in demonstrating its effectiveness across diverse applications and its impact on downstream task performance). The reviewer does not find the evaluation conclusive for this point.
4. The evaluation effectively demonstrates the superiority of the proposed method in both denoising and reconstructing clipped peaks compared to baseline approaches. However, the practical implications for real-world applications remain unclear. Specifically, it is uncertain how much improvement in downstream performance (.e.g, in navigation, localization, or sensor fusion) denoising+reconstruction actually delivers. Evaluating the method within such end-to-end pipelines would strengthen the paper's claims and its practical utility.

---

> ### Author Rebuttal · Authors · 2025-07-30
>
> Thank you for your constructive comments and suggestions. Next is our response.
>
> ## Weakness1: On the Either-Or Routing Choice.
>
> **A1.** We thank you for your in-depth thinking and professional questions regarding our routing logic. This seems to be due to our unclear wording. In fact, our routing mechanism can ensure the use of two experts simultaneously, the use of a single expert, or even no expert at all.
>
> Our framework is designed to handle mixed signals effectively. According to Algorithm 1, when a signal segment meets both clipping and noise conditions, it is routed to both the Over-Range Reconstruction Expert and the Denoise Expert. The final output is then constructed through an intelligent fusion strategy: reconstructed results from the Over-Range Reconstruction Expert are used for the clipped, saturated parts of the signal, while denoised results from the Denoise Expert are applied to the non-saturated, noisy parts. This region-specific processing approach allows efficient handling of mixed-signal scenarios as described.
>
> We conducted a supplementary experiment using mixed-signal test samples, created by combining real over-range events with real noise segments, to quantitatively evaluate our framework’s performance on complex signals.
>
> | Signal Type     | PSNR (↑) | SNR (↑)  |
> | :-------------- | :------- | :------- |
> | Over-range Only | 8.29 dB  | N/A      |
> | Noise Only      | N/A      | 24.19 dB |
> | Mixed Signal    | 7.92 dB  | 22.89 dB |
>
> The results show our MoE-Gyro framework maintains high reconstruction and denoising performance on mixed signals. We will include this experiment and analysis in the appendix and improve the explanation of our MoE approach in the final manuscript.
>
> ## Weakness2: On the Heuristic-based Gating Mechanism.
>
> **A2.** We thank you for your professional feedback on our gating mechanism design. Your points regarding a learned gate and confidence estimation are indeed important directions for improving the system's robustness and intelligence. Our choice of a rule-based heuristic gate was a trade-off, with its main advantages being: near-zero computational overhead, as well as clear and predictable behavior, which greatly facilitates system debugging, validation, and deployment.
>
> We considered using a lightweight MLP as a learned gate for classifying pure noise, pure over-range, and mixed signals. However, since these signal types have distinct features like amplitude and saturation duration, a simple heuristic gate sufficed for current needs. We supplemented the training with an MLP as a pre-classification gate and conducted comparative experiments on mixed signals. The experimental results show that MLP has a strong ability to identify noise in mixed signals, while overrange experts are sometimes not called upon.
>
> | Gate Type      | PSNR(↑) | SNR(↑)   |
> | :------------- | :------ | :------- |
> | Heuristic gate | 7.92 dB | 22.89 dB |
> | MLP gate       | 7.65 dB | 23.91 dB |
>
> We agree that for more complex cases, a learned gate would be better, and we plan to explore such neural network gate architectures as a key focus in future work.
>
> ## Weakness3: On the Downstream Impact of PINN.
>
> **A3.** Thank you for your insightful comments and request for stronger evidence on our PINN design. We agree that its value requires robust cross-scenario validation and have conducted an ablation study to quantitatively assess PINN’s role across diverse dynamic tasks.
>
> We selected three representative tasks with distinctly different dynamic characteristics from our dataset and compared the performance of our full model (with PINN) against a version with the PINN module removed (without PINN). The selected task categories include periodic swings (running), impacts (jumping), and high-frequency twists (wrist).
>
> | Task Type | PSNR (with PINN)(↑) | PSNR(without PINN)(↑) |
> | :--- | :--- | :--- |
> | Running Swing | 8.33 dB | 8.48 dB |
> | Jumping Impact | 7.70 dB | 7.39 dB |
> | Wrist Twist | 8.18 dB | 8.01 dB |
>
> For the simple Running Swing task, the unconstrained model might achieve a slightly higher score, potentially through overfitting. However, looking at the overall picture, the performance of the PINN-constrained model is far more stable and consistent than the unconstrained one. Its performance does not fluctuate drastically with significant changes in task dynamics.
>
> This precisely demonstrates the core role of PINN as a regularizer: by introducing a physical prior, it trades a small amount of overfitting capability on simple tasks for significantly stronger robustness and generalization capability on complex and diverse tasks.
>
> ## Weakness4： On the Improvement in Downstream Tasks.
>
> **A4.** Thank you for highlighting this important limitation. We fully agree that end-to-end downstream evaluation is the ultimate validation of our work. The absence of such studies in our current paper is primarily because our focus has been on tackling the foundational challenge of enhancing MEMS gyroscope performance at the signal-source level. Investigating downstream tasks will be one of our key directions for future research.
>
> Furthermore, we would like to clarify that the core metrics we improved, especially the Allan Variance indicators, are mathematically and empirically linked to downstream navigation performance[1]. In the field of inertial navigation, established theory and extensive literature have proven that a sensor's Bias Instability (BI) is the primary cause of integrated position error that grows quadratically or cubically with time [2]. Meanwhile, Angle Random Walk (ARW) leads to error growth proportional to time to the power of 1.5 [2]. Therefore, the substantial reduction of these metrics in our work will, in theory, necessarily suppress long-term navigation drift, leading to significant accuracy improvements in downstream tasks such as navigation and localization.
>
> Thank you for your valuable suggestion. Integrating our framework into a full navigation system and evaluating its impact on final trajectory accuracy is a key direction of our future work. Your feedback has reinforced our commitment to this goal, and we will highlight it clearly in the future work section of our paper.
>
> ## Limitation1： Discussion of Alternative MoE Architectures.
>
> **A5.** We wholeheartedly agree that including a discussion of alternative MoE architectures would greatly enrich our paper, offer valuable insights for future researchers, and broaden its overall impact. While some related points are touched on in A2 of the rebuttal, we will add a dedicated subsection in the final manuscript to explore this topic in more depth.
>
> We hope the revised version better meets your expectations, if you can improve the score then we appreciate it.
>
> **Reference：**
>
> [1] K. Yuan and Z. J. Wang, “A simple self-supervised imu denoising method for inertial aided
>  navigation,” IEEE Robotics and Automation Letters, vol. 8, no. 2, pp. 944–950, 2023.
>
> [2] Woodman O J , Woodman C O J .An introduction to inertial navigation.Journal of Navigation, 2007, 9(3).

---

> > ### Comment · Reviewer_Ztva · 2025-08-04
> >
> > Thank you for your response and I appreciate the authors' efforts to address the concerns. While some of my questions were addressed satisfactorily, others will likely need to be resolved through future work. Given this, I believe my original score still accurately reflects the current state of the submission, and I will be maintaining it.

---

> > > ### Author Response · Authors · 2025-08-05
> > >
> > > We sincerely appreciate your response and the time you dedicated to rewiewing our paper. Your valuable suggestions have been immensely helpful, and we will incorporate these into the revised version of our paper. We are also pleased to have addressed the concerns you raised about our work. Thank you once again for your support and guidance.

---

### Official Review · Reviewer_TRuX · 2025-07-03

**Clarity:** 4
**Significance:** 3
**Originality:** 3
**Rating:** 5
**Confidence:** 4

**Summary:**

This paper aims to achieve long-range measurement and denoising performance at the same time. It adopts a Mixture of Expert (MoE) structure consisting of two experts, one for long range reconstruction and one for denoising. This method achieves state of the art (SoTA) performance on their proposed dataset called ISEBench.

**Questions:**

1 During training of denoise expert (DE), why can we assume that the randomly sample real recording clip s(t) is noise-free? If not, how can you make sure that DE cancels the noise in s(t)?

2 From Table 5, we can see that the contribution of PINN is relatively small. Is it because the ideal mass–spring dynamics may not be valid in practical cases?

**Ethical Concerns:**

["NO or VERY MINOR ethics concerns only"]

**Final Justification:**

During rebuttal discussions, the authors satisfactorily addressed my main concerns about the following issues.
1. the tradeoff between computation cost and effectiveness, through additional experiments on PSNR, SNR against inference time compared with other baselines.
2. Across device and across task generalization, through additional experiments.
3. the realism of the synthetic noise clarified my concerns on the implicit supervision.
That's why I raised my score from 4 to 5.

**Limitations:**

See weaknesses.

**Paper Formatting Concerns:**

No major formatting issues.

**Quality:**

3

**Strengths And Weaknesses:**

Strengths:
1 The paper is in general well written, with easy-to-follow language.
2 The first self-supervised method to tackle both over range measurement and denoising together.
3 Provide an open source IMU signal enhancement test bench.

Weaknesses:
1 The paper mentioned that the main weaknesses of traditional model-based methods is on the increased power draw and manufacturing costs. However, the proposed self-supervised architecture is also relatively complex and computationally heavy.
2 The training and test split across datasets isn’t clearly described in terms of temporal or domain diversity. Namely, there’s no cross-device or cross-domain generalization test.
3 The paper claims that it is doing label-free self supervised learning. However, when training the denoise expert (DE), it make use of synthetically generated noise, which may introduce implicit supervision from the noise assumptions.

---

> ### Author Rebuttal · Authors · 2025-07-30
>
> Thank you for your constructive comments and suggestions. Next is our response.
>
> ## Weakness1. On the Issue of Cost-Effectiveness.
> **A1.** We fully agree that computational cost is a critical factor for software solutions addressing hardware limitations. We explain below, from three perspectives, why our method offers strong cost-effectiveness.
>
> **Algorithmic Efficiency Optimization and Trade-offs.** Through pruning, we have successfully compressed our model to approximately 1.85M, a size that is well within the capacity of modern embedded AI platforms. To intuitively demonstrate its benefits, we provide the following performance-efficiency table（Inference time measured on RTX4060）:
>
> | Model | PSNR (dB) (↑) | SNR (dB) (↑) | Inference Time (ms) (↓) |
> | :--- | :--- | :--- | :--- |
> | MoE-Gyro (Full) | 8.29 | 24.19 | 117 |
> | MoE-Gyro (Comp.) | 7.89 | 21.60 | 41 |
> | Heros_GAN | 7.70 | 16.86 | 39 |
> | IMUDB | 6.59 | 17.76 | 173 |
>
>
> The table shows our compressed model greatly improves efficiency with minimal performance loss, supporting low-cost deployment. Theoretically, our proposed MoE decoupling framework can accommodate models of any size, allowing smaller models to be used for edge deployment.
>
> **Synergy with Hardware Process Improvements.**  Our framework complements hardware advances; its decoupling design supports all sensor levels and delivers better results as sensor quality improves.
>
> **System-Level Computational Cost(Reusing Existing Compute Units).**  Our method offers a substantial advantage in overall system cost by eliminating the need for dedicated compute hardware. Traditional solutions often depend on complex ASICs and high-precision manufacturing, which increase hardware expenses. In contrast, our algorithm can be deployed on existing compute units already integrated into platforms such as UAVs and robots, where they are typically used for tasks like vision, planning, and control. By running as an additional task, our method introduces minimal computational overhead, requiring no extra hardware or power consumption.
>
> In summary, through its own algorithmic optimization, synergy with hardware advancements, and the reuse of computational resources at the system level, our method provides a more flexible and cost-effective solution than the traditional hardware upgrade path.
>
> ## Weakness2: Generalization Capability.
>
> **A2.** This important question highlights our model’s practical utility. We will clarify how our framework adapts to different IMUs and present two experiments demonstrating its generalization across devices and tasks.
>
> **Device:** Before presenting the experimental results, we would like to briefly clarify how our framework adapts to different IMU models. The process is straightforward: the user only needs to adjust the input signal to 100Hz via standard downsampling or interpolation and ensure the units are in rad/s.
>
> To quantitatively evaluate the model's generalization to different hardware, we used the model trained on ISEBench (with data primarily from an iPhone 14) to perform zero-shot generalization testing on new data from a Huawei Pura 70 and a Xiaomi 14. The results are as follows:
>
> | Device | PSNR (MoE-Gyro) (↑) | PSNR (HerosGAN) (↑) | ARW Red. (MoE-Gyro) (↑) | ARW Red. (HerosGAN) (↑) |
> | :--- | :--- | :--- | :--- | :--- |
> | Training Device (iPhone14/Bosch BMI160) | 8.29 dB | 7.70 dB | 94.10% | 51.60% |
> | Xiaomi 14 (Zero-shot) | 7.95 dB | 7.51 dB | 86.50% | 48.30% |
> | Huawei Pura 70 (Zero-shot) | 8.28 dB | 7.41 dB | 88.90% | 45.70% |
>
> The results clearly indicate that on the over-range reconstruction task (measured by PSNR), our model exhibits excellent zero-shot generalization capabilities, with almost no performance loss on the Huawei device and maintaining a high standard on the Xiaomi device. On the denoising task (measured by the ARW reduction percentage), the zero-shot performance shows a slight decrease. This is expected, as the baseline noise characteristics differ across devices. Further results after fast fine-tuning will be included in the final manuscript.
>
> **Task:**  To validate the model's generalization to different types of dynamic tasks, we categorized and evaluated its performance on over-range signals from our dataset based on their motion cause, including periodic swings (running), impacts (jumping), and high-frequency twists.
>
> | Task Type | PSNR (MoE-Gyro) (↑) | PSNR (HerosGAN) (↑) |
> | :--- | :--- | :--- |
> | Running Swing | 8.33 dB | 7.04 dB |
> | Jumping Impact | 7.70 dB | 7.73 dB |
> | Wrist Twist | 8.18 dB | 7.56 dB |
>
> The results confirm our model's stable, high performance on diverse dynamic tasks, suggesting it captures general reconstruction principles rather than overfitting to motion patterns.
>
> Therefore, the strong cross-device and cross-task results suggest that the MoE-Gyro framework has robust generalization and broad applicability in real-world scenarios.
>
> ## Weakness3: On the Issue of Implicit Supervision.
> **A3.** We thank you for your insightful point regarding implicit supervision. Although data augmentation introduces prior knowledge, our approach retains key advantages over supervised methods, and we provide justification for using predefined noise types.
>
> **Advantage:** Traditional supervised learning needs costly, high-precision sensors with perfect synchronization for ground-truth data. Our self-supervised method avoids these expenses, which is why we call it label-free self-supervised learning.
>
> We want to emphasize that for sensors like MEMS gyroscopes, the primary random noise processes are not infinitely unknown. Based on decades of research and industry standards such as those from the IEEE, the noise characteristics are dominated by a finite set of key components with well-defined physical models[1]. In other words, any real-world noise is a combination of these noise types.
>
> As described in Section 4.2 of our paper, these noise components can be precisely identified and quantified using Allan Variance, and they primarily include Quantization Noise (QN), Angle Random Walk (ARW), and Bias Instability (BI). These noises have distinct spectral characteristics. Therefore, our generation method uses combinations of these noise types to simulate real-world noise. This makes our synthesis task both reasonable and bounded.
>
> In short, our FFT-guided noise synthesis uses real sensor PSD data instead of assuming simple Gaussian noise, providing a data-driven weak prior rather than relying on labeled signals from external devices.
>
>
> ## Question1: On the Noise within the Real Signal $s(t)$ during DE Training
>
> **A1.** Thank you for the insightful question. We do not assume that $s(t)$ is noise-free. Our design focuses on enabling the model to learn a denoising mapping by training it to remove the extra synthetic noise $x_{noise}^{\prime}(t)$ that we inject. This comparative training approach allows effective denoising even when $s(t)$ contains noise.
>
> * **Training Input :**
>     $$x_{mix} = (x_{noise}(t) + \alpha s(t)) + x_{noise}^{\prime}(t) $$
> * **Training Target :**
>     $$x_{clean} = x_{noise}(t) + \alpha s(t)$$
>
> The model preserves the original noise because it is included in the training target $x_{clean}$. To minimize loss, the model must fully reproduce the target signal, including baseline noise from both $x_{noise}(t)$ and $s(t)$.
>
> At a practical level, we use the scaling factor $\alpha$ to adjust the scale of $s(t)$ to be just slightly above the baseline noise level. This ensures that the absolute energy of the noise contained within $s(t)$ is kept at a low level (much lower than the synthetic noise $x_{noise}^{\prime}(t)$), preventing it from becoming a dominant interference in the training process.
>
> ## Question2: The Impact of the PINN.
> **A5.** We thank you for your detailed analysis of our ablation study. The addition of PINN to the final model does bring a certain level of improvement. Furthermore, regarding the positioning of PINN, we see it primarily as a physics-based regularizer. We hope to use it to:
> - **Ensure physical plausibility:** making sure the reconstructed signal waveform adheres to basic dynamic principles and avoids unrealistic, overly-oscillatory, or overly-smoothed pseudo-signals.
> - **Enhance generalization capability:** by incorporating universal physical laws, it guides the model to learn more general features, thus making it more robust when encountering new motion patterns not seen in the training set.
>
> Our PINN loss is based on an idealized physical model. As described in Appendix B of our paper, we calculate the discrete specific power, $e_t=(\Delta^{2}x_{t-1}+\Delta^{2}x_{t})\Delta x_t$, and then use a barrier loss function to penalize physically implausible reconstructions that have either excessively high (oscillatory) or low (overly-smoothed) energy.
>
> However, real-world sensors show complex effects beyond our ideal mass-spring model, which may limit PINN’s performance gains. Nonetheless, PINN helps ensure physical plausibility and improves generalization.
>
> We hope the revised version better meets your expectations, if you can improve the score then we appreciate it.
>
> **Reference：**
>
> [1]IEEE Standard Specification Format Guide and Test Procedure for Single-Axis Laser Gyros," in IEEE Std 647-2006 (Revision of IEEE Std 647-1995) , vol., no., pp.1-96, 18 Sept. 2006.

---

> > ### Comment · Reviewer_TRuX · 2025-08-03
> >
> > Thank you to the authors for the comprehensive rebuttal. The rebuttal satisfactorily addresses most of my earlier concerns.
> >
> > A follow-up question about Q1: Since the real signal s(t) may still contain noise, does the method assume that this residual noise is acceptable in downstream applications? Or do you envision an additional stage that further denoises s(t) itself?

---

> > > ### Author Response · Authors · 2025-08-04
> > > **Response for follow-up question about Q1**
> > >
> > > **A1.** Thank you for this insightful follow-up question. It has prompted us to think more deeply about the nature of our training paradigm and its future possibilities.
> > >
> > > Our core design philosophy is indeed to learn a denoising mapping through a comparative process, from a "high-noise" state ($x_{mix}$) to a "low-noise" state ($x_{clean}$), by training the model to precisely remove the extra injected noise, $x_{noise}^{\prime}(t)$. Within this framework, you are correct that the lower the inherent baseline noise in $s(t)$, the more ideal our training target $x_{clean}$ becomes, which could theoretically lead to a higher performance ceiling for the model.
> > >
> > > **Your question has inspired a very valuable extension of our thinking:**
> > >
> > > We could use our existing MoE-Gyro framework to pre-process our repository of weak motion signals, $s(t)$. Specifically, we could:
> > >
> > > 1.  First, use the currently trained model to perform an initial denoising pass on all $s(t)$ segments, creating a "cleaner" signal repository, $s_{clean}(t)$.
> > > 2.  Then, use this cleaner $s_{clean}(t)$ to repeat our Denoise Expert (DE) training process.
> > >
> > > This iterative training or self-purification approach could, in theory, further enhance the model's denoising performance.
> > >
> > > We sincerely thank you for this suggestion, it has illuminated a very interesting and promising path for iterative optimization. We will add a discussion of this iterative optimization approach to the "Discussion" section of our final manuscript to enrich our work and inspire subsequent research.

---

> > > > ### Comment · Reviewer_TRuX · 2025-08-05
> > > >
> > > > Thanks for the clarification on my follow-up question. That iterative denoising idea reminds me of [1], which is a very interesting future extension.
> > > > I will gladly raise my score to 5.
> > > >
> > > > [1] Revach, Guy, et al. "RTSNet: Learning to smooth in partially known state-space models." IEEE Transactions on Signal Processing 71 (2023): 4441-4456.

---

> > > > > ### Author Response · Authors · 2025-08-05
> > > > >
> > > > > Thank you very much for your recognition and support, and for raising our score to 5!
> > > > >
> > > > > The reference you provided on iterative denoising is highly insightful, and we appreciate you bringing this excellent related work to our attention. We will study it carefully and will incorporate it into our discussion of the iterative optimization approach in the "Future Work" section of the final manuscript.
> > > > >
> > > > > Thank you once again for your invaluable feedback and profound insights throughout the entire review process!

---

### Official Review · Reviewer_io28 · 2025-07-05

**Clarity:** 3
**Significance:** 3
**Originality:** 3
**Rating:** 4
**Confidence:** 2

**Summary:**

This paper investigates the fundamental trade-off between measurement range and noise performance in MEMS gyroscopes: a high measurement range often comes with high noise, while low noise usually limits the range. The authors propose MoE-Gyro, which simultaneously achieves over-range signal reconstruction and noise suppression without adding extra hardware complexity. They insightfully recognize that directly using a single network to learn both “reconstructing saturated signals” and “signal denoising” would encounter gradient imbalance problems, and thus cleverly incorporate task decoupling, designing loss functions and data augmentation strategies specifically for the two modules. Experimental results demonstrate the advantages of this method over existing technologies, notably achieving substantial sensor performance improvements under unsupervised conditions. Additionally, by releasing datasets and benchmark tools, the authors make a positive contribution to advancing research in this field.

**Questions:**

NA.

**Ethical Concerns:**

["NO or VERY MINOR ethics concerns only"]

**Limitations:**

Yes.

**Paper Formatting Concerns:**

No.

**Quality:**

3

**Strengths And Weaknesses:**

Strengths:
1. This paper proposes a novel unified framework that simultaneously addresses the problems of over-range reconstruction and denoising for MEMS gyroscopes. The authors introduce a mixture-of-experts model into the domain of IMU signal enhancement. Each module is well designed to align with the task characteristics, effectively tackling these challenges.
2. The overall architecture follows a decoupling philosophy, employing two dedicated models to separately handle the reconstruction and denoising tasks, thus avoiding mutual interference between the multitasks. The gated routing strategy is simple and efficient. The overall algorithm is solidly designed and works smoothly.
3. On the ISEBench benchmark constructed by the authors, MoE-Gyro achieves state-of-the-art performance across the metrics, showing significant improvements over traditional filtering and existing deep learning methods.
4. The authors have also released a high-quality dataset and evaluation toolkit: the IMU Signal Enhancement Benchmark, which I believe will benefit the community.
5. Additionally, the paper is clearly written and provides detailed methodological descriptions, from which I could effectively gain information.

Weaknesses:
1. The paper targets applications such as unmanned aerial vehicles and robotics. Thus, algorithm efficiency should also be an important evaluation metric. However, the experiments seem to lack efficiency comparisons. Is it real-time on a PC (RTX4060)? And can it achieve real-time performance on embedded platforms for real-world applications?
2. More of a question than a weakness: the paper has been thoroughly tested on ISEBench, which includes three datasets. I would like to know how MoE-Gyro generalizes to IMU devices beyond ISEBench. Does it require fine-tuning for specific IMUs?

---

> ### Author Rebuttal · Authors · 2025-07-30
>
> Thank you for your constructive comments and suggestions. Next is our response.
>
>
> ## Weakness1: Efficiency and Real-Time Performance.
>
>
> **A1.** We completely agree that for applications such as UAVs and robotics, efficiency is a core metric of equal importance to accuracy.  To address this, we evaluated our method in terms of real-time PC performance, embedded deployment feasibility, and its impact on downstream systems.
>
> First, it is important to note that a key innovation of our work is decoupling denoising and reconstruction using an MoE framework, allowing flexible, pluggable expert models. In theory, all lightweight architectures like CNNs or MLPs can be used to adapt the framework for edge deployment.
>
> **Real-time Performance on PC:** Our full MoE-Gyro model, tested on an NVIDIA RTX 4060, processes 2.56 seconds of IMU data (256 samples @ 100Hz) in just 0.117 seconds. In a practical application, this means we can use an overlapping sliding window strategy with a 0.5-second step size to achieve a 2Hz (0.5s) data update rate, which meets the requirements of many online applications that demand high refresh rates.Theoretically, the maximum update rate of the model is 8Hz, and the maximum update rate of the compressed model is 24Hz.
>
> **Analysis for Embedded Deployment:** To address embedded deployment concerns, we compressed our model to about 1.85M, reducing parameters by 12×. This size is comparable to lightweight models like MobileNetV2, showing strong potential for deployment on embedded AI platforms such as NVIDIA Jetson after standard optimization. Then, we compared the performance and efficiency of our full and compressed models with baselines to illustrate trade-offs intuitively.（Inference time measured on RTX4060）
>
> | Model | PSNR (dB) (↑) | SNR (dB) (↑) | Inference Time (ms) (↓) |
> | :--- | :--- | :--- | :--- |
> | MoE-Gyro (Full) | 8.29 | 24.19 | 117 |
> | MoE-Gyro (Comp.) | 7.89 | 21.60 | 41 |
> | Heros_GAN | 7.70 | 16.86 | 39 |
> | IMUDB | 6.59 | 17.76 | 173 |
>
>
> The table shows our compressed model greatly improves efficiency with only a slight performance drop, outperforming other methods in the performance-efficiency trade-off. Smaller models can also be used by replacing Transformer backbones with CNNs or MLPs for specific tasks.
>
> **Role in Downstream Systems:** Finally, we wish to clarify the precise role of our algorithm within a robotic or UAV system. It is not intended to replace the low-level attitude control loop that runs at several hundred Hertz. Instead, MoE-Gyro acts as a Navigation Co-processor. At a moderate frequency (e.g., 2Hz), it provides the system's mid-level state estimator with a motion update that has been denoised and over-range reconstructed. This is crucial for ensuring long-term navigation accuracy in GPS-denied environments.
>
> ## Weakness2:  Generalization Capability.
>
>
> **A2.** This important question highlights our model’s practical utility. We will clarify how our framework adapts to different IMUs and present two supplementary experiments demonstrating its generalization across devices and tasks.
>
> **Device:**  We only need to resample the signal frequency to adapt to any IMU device. Our framework accepts signals at a 100Hz input rate. Therefore, for any new IMU, the user simply needs to adjust the signal to 100Hz via standard downsampling or interpolation before feeding it to the model. Attention must also be paid to the units of the angular velocity signal; if the device's output is in deg/s, a unit conversion to rad/s (by multiplying by π/180) is required. The framework has a built-in Z-score normalization module, so users do not need to manually handle data scaling issues.
>
> To quantitatively evaluate the model's generalization to different hardware, we conducted a cross-device experiment. We used the model trained on the ISEBench dataset and performed zero-shot generalization testing on new data collected from a Huawei Pura 70 and a Xiaomi 14.
>
> | Device | PSNR (MoE-Gyro) (↑) | PSNR (HerosGAN) (↑) | ARW Red. (MoE-Gyro) (↑) | ARW Red. (HerosGAN) (↑) |
> | :--- | :--- | :--- | :--- | :--- |
> | Training Device (iPhone14/Bosch BMI160) | 8.29 dB | 7.70 dB | 94.10% | 51.60% |
> | Xiaomi 14 (Zero-shot) | 7.95 dB | 7.51 dB | 86.50% | 48.30% |
> | Huawei Pura 70 (Zero-shot) | 8.28 dB | 7.41 dB | 88.90% | 45.70% |
>
> The results show our model generalizes well to new devices, with minimal drop in over-range reconstruction and slight decline in denoising. More experiments will be included in the final manuscript.
>
> **Task:**  To verify whether the model can generalize to different types of dynamic tasks, we performed a simple classification of the over-range signals in our data based on their motion cause and evaluated the model's performance on each. The main categories include: periodic swings (running), impacts (jumping), and high-frequency twists. The performance is as follows:
>
> | Task Type | PSNR (MoE-Gyro) (↑) | PSNR (HerosGAN) (↑) |
> | :--- | :--- | :--- |
> | Running Swing | 8.33 dB | 7.04 dB |
> | Jumping Impact | 7.70 dB | 7.73 dB |
> | Wrist Twist | 8.18 dB | 7.56 dB |
>
> The experimental results show that our model's performance is stable and remains at a high level when processing these distinctly different dynamic tasks. This proves that it has learned the general principles of signal reconstruction, rather than memorizing specific motion patterns.
>
> Therefore, based on the strong zero-shot cross-device performance and the stable cross-task performance, we are confident that the MoE-Gyro framework possesses excellent generalization capabilities and can adapt to diverse real-world application scenarios.
>
> We hope the revised version better meets your expectations, if you can improve the score then we appreciate it.

---

> > ### Comment · Reviewer_io28 · 2025-08-09
> >
> > After reviewing the other reviewers' comments and the authors' rebuttal, I think the paper meets NeurIPS's acceptance bar.

---

> > > ### Author Response · Authors · 2025-08-09
> > >
> > > Thank you very much for your time and positive feedback during the discussion period. We are very pleased that our rebuttal was well-received, and we sincerely appreciate your support for our work!

---

### Official Review · Reviewer_gDzY · 2025-07-06

**Clarity:** 3
**Significance:** 3
**Originality:** 3
**Rating:** 4
**Confidence:** 3

**Summary:**

This paper introduces MoE-Gyro, a self-supervised Mixture of Experts (MoE) framework, for simultaneous over-range signal reconstruction and noise suppression in MEMS gyroscopes. It features an Over-Range Reconstruction Expert (ORE) utilizing Gaussian-Decay Attention and physics-informed constraints, and a Denoise Expert (DE) with dual-branch complementary masking and FFT-guided augmentation. The paper conducts experiments on three public datasets and introduced a benchmark named ISEBench.

**Questions:**

1. While task decoupling indeed resolves the gradient conflict, it might also mean that subtle synergistic effects between certain tasks are overlooked. For example, it's worth considering whether very high noise levels might change the nature of an over-range signal, or if certain over-range signals are, in fact, noise-induced.

2. It's questionable whether the "a gate then applies a simple heuristic to route each segment to..." approach can adequately handle complex noise scenarios. For instance, how would it perform when a signal segment simultaneously contains both over-range signals and noise?

3. The paper's "FFT-guided training augmentation" method, designed to enhance data by combining noise PSD with weak motion signals, lacks a discussion of its limitations in synthesizing data that accurately reflects complex real-world noise conditions.

4. The paper contains several typographical errors. For instance, on line 54, "Over-Range Reconstruction Expert (DE)" should be corrected to "Denoise Expert (DE)."

**Ethical Concerns:**

["NO or VERY MINOR ethics concerns only"]

**Final Justification:**

I would like to thank the authors for their detailed response to my questions. My major concerns have been satisfactorily addressed. Therefore, I decide to raise my rating to weak accept.

**Limitations:**

Yes

**Quality:**

3

**Strengths And Weaknesses:**

Strengths:

The paper proposes a novel approach to IMU precise navigation by decoupling over-range signal reconstruction and noise elimination into two self-supervised frameworks. The method seems novel. Additionally, the experimental design and provided code are comprehensive. The writing is clear, with no major issues.

Weaknesses:

1. The problem studied might be important for MEMS Gyroscopes. But the topic may have limited attention and influence in the NeurIPS community. It would be more appropriate to submit the paper to a conference in more relevant field.

2. Some details of the proposed system framework are not discussed in sufficient depth. The method may exhibit anomalous behavior when encountering complex or unexpected IMU signals, and the paper lacks analysis and experimental validation in this regard.

---

> ### Author Rebuttal · Authors · 2025-07-30
>
> Thank you for your constructive comments and suggestions. Next is our response.
>
> ## Weakness1: Relevance and Impact within the NeurIPS Community.
>
> **A1.** Thank you for your thoughtful feedback. We believe our paper exemplifies the growing field of AI for Science (AI4Sci), which is gaining momentum within the NeurIPS community. Our key contributions include:
> - **AI Empowerment for Overcoming Fundamental Scientific Bottlenecks:** Our work tackles a long-standing challenge in sensor physics, namely the trade-off between measurement range and noise in MEMS gyroscopes. Traditional hardware approaches are constrained by the high cost and complexity of manufacturing and fabrication. We show that AI can overcome these physical limitations through a software-based approach, providing a novel solution for scientific instrument design and expanding AI's role in scientific applications.
> - **Creating New Tools for Scientific Exploration:** Our MoE-Gyro framework boosts low-cost consumer-grade sensors to near tactical-grade performance by AI, making high-precision motion sensing accessible to resource-limited labs, startups, and researchers. This enables scientific exploration in fields such as robotics, autonomous driving, and motion tracking, which previously required expensive equipment.
> - **Building an Open Benchmark to Drive Field-Wide Progress:** We recognize that progress in AI4Sci depends on open and reproducible research. To support this, we introduce ISEBench, the first open-source benchmark for IMU signal enhancement, providing a fair evaluation platform to advance AI-driven inertial sensing research.
>
> In conclusion, our study falls within the AI4Sci domain and aligns well with NeurIPS’s research focus. We believe it offers meaningful contributions both to the MEMS field and to the broader development of AI methodologies.
>
>
> ## Weakness2: Performance under complex or unexpected IMU signals.
> **A2.** Thank you for raising this question. We have addressed it in A3 and A4 of this rebuttal. A more in-depth analysis and additional experiments will be included in the next version of the paper.
>
> ## Question1: On the Synergistic Effects of Over-Range Signals and Noise.
> **A3.** Thank you for this insightful question. We would first like to characterize the physical origins of noise. In our context, noise can be classified into two categories: internal device noise and undesired external inputs from the perspective of a downstream task[1].
> - **Internal Device Noise:** This noise primarily originates from thermo-mechanical and electronic sources within the sensor, defining its baseline noise floor. Its amplitude is typically far below the sensor's full-scale range and thus cannot induce an over-range event on its own. Our Denoise Expert (DE) is specifically trained to efficiently filter this type of noise and output a clean signal.
> - **Undesired High-Amplitude External Inputs:** As you rightly pointed out, some over-range signals can indeed be considered as being caused by task-level noise, such as a mechanical shock experienced by a UAV or severe jolts in a vehicle. To the sensor, these external disturbances are legitimate physical inputs with amplitudes sufficient to cause saturation. This results in a complex mixed signal, which contains both clipped peaks that need reconstruction and potentially post-shock ringing noise that needs to be filtered. Our MoE architecture is designed to decouple and process this signal: the Over-Range Reconstruction Expert (ORE) is responsible for precisely recovering the waveform of the saturated parts. The Denoise Expert (DE) is responsible for cleaning up the noise in the non-saturated parts, and a Gate mechanism to route signals to the appropriate expert.
>
> Therefore, whether handling low-level internal noise or complex mixed signals caused by high-intensity external disturbances, our MoE framework demonstrates the feasibility and superiority of its design. By decoupling the tasks of reconstruction and denoising, which have vastly different mathematical objectives. This provides high-quality, wide-range, low-noise signals that form a reliable foundation for improving downstream tasks, including single-sensor and multi-sensor fusion applications.
>
> ## Question2: Performance on Handling Complex Signals.
>
> **A4.** We thank you for your concern regarding the robustness of our gating mechanism. To comprehensively address your doubts, we will first elaborate on our gating design philosophy and then present the results of a supplementary experiment specifically conducted to validate its performance on complex signals.
>
> Our gating is designed to balance efficiency and practicality by using a lightweight, rule-based heuristic with minimal computational cost, suitable for real-time processing. Its main role is to quickly and accurately decide whether a signal segment requires reconstruction, denoising, or both.
> Specifically, according to Algorithm 1, when consecutive saturation points are detected, a peak flag is set; if the amplitude remains below a preset threshold $\tau$ for a continuous period, a noise flag is also set. If both flags are true, the segment is sent to both the Over-Range Reconstruction Expert (ORE) and the Denoise Expert (DE) simultaneously, enabling region-specific processing. The final output fuses both experts’ results: ORE’s reconstruction is used for saturated parts, while DE’s denoised output is used for non-saturated parts, ensuring fine-grained handling of complex signals.
>
> To further validate our gating and fusion strategy, we conducted a supplementary experiment focusing on complex signal segments with both over-range and noise components, as you mentioned, confirming the effectiveness of our approach.
> - Experimental Setup: We created 1,000 mixed-signal test samples by combining real over-range events from the GyroPeak-100 dataset with realistic noise segments spliced into non-saturated regions, ensuring complex and authentic test data.
> - Evaluation Metrics: For mixed signals, we used two metrics: Peak Signal-to-Noise Ratio(PSNR) to assess reconstruction accuracy in saturated parts and Signal-to-Noise Ratio(SNR) to evaluate denoising effectiveness in non-saturated noisy parts.
>
> | Signal Type | PSNR (↑) | SNR (↑) |
> | :--- | :--- | :--- |
> | Over-range Only | 8.29 dB | N/A |
> | Noise Only | N/A | 24.19 dB |
> | Mixed Signal | 7.92 dB | 22.89 dB |
>
> The experimental results clearly show that even when processing mixed signals, the reconstruction and denoising performance of our MoE-Gyro framework experiences only a slight decrease.  This provides strong evidence that our gating mechanism and subsequent fusion strategy are both robust and efficient. We will add this supplementary experiment and its analysis to the appendix of our final manuscript to enhance the rigor of our work.
>
> ## Question3:  The Limitations of Synthetic Noise.
>
> **A5.** Thank you for your concern about the authenticity Tof our data augmentation. In fact, the types of noise present in gyroscopes are known. We start by defining key MEMS gyroscope noise types based on IEEE standards[2] and Allan Variance analysis[3], which identifies distinct spectral noise components through slope fitting on Allan Variance plots, including: Quantization Noise (QN): Primarily affects the high-frequency band and corresponds to white noise in the frequency domain. Angle Random Walk (ARW): The dominant noise in the mid-frequency band, corresponding to white noise in the angular rate domain.Bias Instability (BI): The dominant noise in the low-frequency band, typically caused by flicker noise (1/f noise), which has a higher power spectral density at lower frequencies.
>
> In other words, even the most complex noise can be decomposed into a combination of these types. Then, the core idea of our proposed FFT-guided training augmentation is to specifically synthesize these key noise components, rather than simply injecting generic Gaussian white noise. Our specific steps are as follows:
> 1. Analyze the Real Noise Spectrum: We first perform a Fast Fourier Transform (FFT) on static data segments collected from the real sensor to obtain its Power Spectral Density (PSD). From the PSD curve, we identify the frequency bands that reflect the different noise components such as QN, ARW, and BI.
> 2. Synthesize Noise with Matched Power in Key Bands: We synthesize noise segments with frequency distributions adjusted to closely match the PSD of real noise, ensuring key characteristics like QN, ARW, and BI are well replicated.
> 3. Injection and Training: We inject the realistic synthetic noise into training data, enabling the Denoise Expert (DE) to learn a robust model that handles real sensor noise. As shown in Figure 5c, this approach significantly outperforms no augmentation or simple Gaussian noise.
> Modeling more complex noise sources remains a valuable direction for future research, and we will add a discussion of this to our revised manuscript to make our work's description more comprehensive.
>
> ## Question4:  Typographical Error.
>
> **A6.** Thank you for your careful review. We will correct this issue and ensure typos are eliminated in the next version. We apologize for the oversight.
>
> We hope the revised version better meets your expectations, if you can improve the score then we appreciate it.
>
> **Reference:**
>
> [1]Fraden,Jacob.Handbook of Modern Sensors[J].Springer New York, 2010, 10.1007/978-1-4419-6466-3.
>
> [2]IEEE Standard Specification Format Guide and Test Procedure for Single-Axis Laser Gyros," in IEEE Std 647-2006 (Revision of IEEE Std 647-1995) , vol., no., pp.1-96, 18 Sept. 2006.
>
> [3]D. Allan, “Statistics of atomic frequency standards,” Proceedings of the IEEE, vol. 54, no. 2, pp.  221–230, 1966.

---

### Note · Authors · 2025-08-12

Dear Area Chair and Reviewers:

We would like to take this opportunity to express our sincere gratitude for your time and the insightful, constructive feedback provided throughout the review process.

Our work is positioned as a compelling case for AI for Science, demonstrating how novel ML architectures can solve long-standing bottlenecks in sensor physics. We are greatly encouraged that the reviewers validated this perspective by reaching a strong consensus on our work's key strengths. To assist the Area Chair in summarizing the outcomes, we have highlighted these main points of agreement below:

**1.A Novel and Well-Motivated Framework (Praised by Reviewer gDzY, io28, TRuX, Ztva)**

**2.A Solid and Thoughtful Design (Praised by Reviewer io28, Ztva)**

**3.A Valuable Contribution to the Community (Praised by Reviewer io28, TRuX, Ztva)**

**4.A Thorough and Convincing Evaluation (Praised by Reviewer gDzY, io28, Ztva)**

The discussion period was incredibly valuable, and based on the reviewers' feedback, we have made the following key clarifications and additions:

**1.Addressed Efficiency and Deployment:** We presented a 12x compressed model that maintains superior performance,demonstrated the model's efficiency.

**2.Validated Generalization Capability:** We supplemented our work with new cross-device and cross-task zero-shot experiments, quantitatively validating our framework's robustness.

**3.Clarified Key Design and Application Details:** We provided detailed explanations on handles mixed signals, clarified the specifics of our noise handling, and defined the framework's practical role in real-world systems.

**The reviewers agree that our rebuttal addresses the concern.**

We are confident that our work presents a significant and well validated contribution that will be of interest to the NeurIPS community. We are fully committed to incorporating all valuable suggestions into our final manuscript to further strengthen the paper.

Thank you again for your time and consideration.

---

### Decision · Program_Chairs · 2025-09-17

**Decision:**

Accept (poster)

**Comment:**

This paper introduces MoE-Gyro, a self-supervised Mixture-of-Experts framework that addresses both over-range reconstruction and denoising for MEMS gyroscopes. The method is well motivated, technically solid, and supported by a thorough evaluation, including a new benchmark (ISEBench) that will be useful to the community. The authors responded well to reviewer concerns, providing results on efficiency, cross-device generalization, and clarifying details on noise modeling and gating. Overall, the paper makes a clear contribution to AI for sensor enhancement and meets the bar for acceptance.